

# Palaeo-landslide dams controlled the formation of Late Quaternary terraces in Diexi, the upper Minjiang River, eastern Tibetan Plateau

Jingjuan Li[1], Xuanmei Fan[1], Zhiyong Ding[1], Marco Lovati[1]

[1] State Key Laboratory of Geohazard Prevention and Geoenvironment Protection, Chengdu University of Technology, Chengdu 610059, China

*Correspondence to*: Xuanmei Fan (fxm_cdut@qq.com)

**Abstract**

Tectonic uplift and climate changes are the two critical factors controlling the evolution of river landscapes and the formation of terraces. However, little is known about the effect of river blockage events on terrace formation along valley areas. In this paper, we investigated the geomorphology, sedimentology, and chronology of Tuanjie (seven staircases) and Taiping (three staircases) Terraces in Diexi. These represent two typical fluvial terraces in the upper Minjiang River in the eastern Tibetan Plateau. These terraces are composed, from bottom to top, of lacustrine deposits, gravels, loess, and paleosol. Taiping Terrace 3 (T3) has two sets of mud-phyllite clasts sequences. Field investigation, Digital Elevation Model (DEM) data, lithofacies, and dating results confirm that terraces T1 to T3 in Taiping correspond to terraces T5 to T7 in Tuanjie. These findings suggest that two damming and four outburst events occurred in the area during the late Pleistocene. The palaeo-dam blocked the river around 32.40±1.91 ka, followed by the first outburst at 27.11±0.18 ka. Then, the palaeo-dam blocked the river again between 27 to 17 ka, and suffered a second dam-breaking event at 17 ka. The third and fourth gradual collapse events respectively occurred at ~10 ka and ~9.35 ka. Combined with the tectonic uplift rate, river incision rate, and high-resolution climate data, our analysis shows that the blockage and collapse of the palaeo-dam have been a major factor in the formation of tectonically active mountainous river terraces. Tectonic movement and climatic fluctuations, on the other end, play a minor role.



## 1 Introduction

Terraces, as the natural archive of the process of valley evolution, are used to explore the evolution and controlling mechanisms of river landscape (Liu et al., 2021; Chen et al., 2020). This landform is sensitive to the impacts of tectonic and climate (Pan et al., 2003; Singh et al., 2017; Do Prado et al., 2022;

Avsin et al., 2019; Gao et al., 2020), it can reflect the dynamics of the fluvial system (Schumm and Parker, 1973), rock uplift rate (Pan et al., 2013; Giano and Giannandrea, 2014; Malatesta et al., 2022), fault activity (Caputo et al., 2008), crustal movement (Westaway and Bridgland, 2007; Yoshikawa et al., 1964; Okuno et al., 2014), glacier melting (Oh et al., 2019; Vásquez et al., 2022; Bell, 2008), sea level (Malatesta et al., 2022; Yoshikawa et al., 1964) and lake level changes (Wang et al., 2021b). In

tectonically active mountainous regions, some extreme events like landslides, debris flows, and rockfalls also change fluvial dynamics and landscape (Molnar et al., 1993; Molnar and Houseman, 2013; Srivastava et al., 2017), thereinto, river blockage and sudden outburst can strongly affect the evolutionary and geomorphology of the upstream and downstream sections (Hewitt et al., 2008; Hewitt et al., 2011; Korup et al., 2007; Korup et al., 2010). At present, there are few studies on the influence of disaster

events on the formation and evolution of terraces, further exploration is advisable.

The rapid uplift and climate change of the Tibetan Plateau in the late Quaternary led to frequent disaster events in its eastern margin (Yang et al., 2021; Dai et al., 2021; Wu et al., 2019; Gorum et al., 2011; Fan et al., 2018; Fan et al., 2017). As a result, the formation factors of river terraces in this region have been controversial, and the causes of the periodicity of the orbital scale (100 ka, 40 ka, 20 ka) and

centennial-scale (0.1 ka) are also unclear.

The upper Minjiang River is located in the eastern Tibetan Plateau, there are wide distribution of the three-level terraces (Yang, 2005). Through these terraces, the development of palaeo-landslide, the variations of climate, and the movement and evolution of regional tectonic uplift have been studied. Due to the incompleteness of relevant data, these studies are still in the exploration phase (Luo et al., 2019;

Yang et al., 2003; Yang, 2005; Zhu, 2014; Gao and Li, 2006).

The terraces in the Diexi area are typical river terraces in the upper Minjiang River, and they are located in the famous Diexi palaeo-dammed lake, which is one of the largest, best-preserved, and longest-duration lakes in a tectonically active mountainous region (Fan et al., 2019). Previous studies found that there are two terraces developed in Tuanjie and Taiping villages (Wang et al., 2005a; Yang et al., 2008;



Fan et al., 2019). The analysis of lithofacies and sedimentary systems determined that the Diexi area is mainly composed of fluvial sedimentary system, lacustrine sedimentary system, alluvial fan sedimentary system, and eolian and supergene sedimentary systems (Yang, 2005; Yang et al., 2008). Unfortunately, the systematic study of the sedimentary facies of the Tuanjie and Taiping Terraces is incomplete. Currently, Tuanjie Terrace is supported to result from the outburst of a palaeo-dammed lake 15000 years

ago, and each terrace may correspond to different stages of outburst (Duan et al., 2002; Wang et al., 2005b; Wang, 2009; Zhu, 2014). That is, the Diexi palaeo-dammed lake has experienced at least one outburst flood event (Ma et al., 2018; Wang et al., 2012; Wang et al., 2005b). However, due to the lack of sedimentary sequence and chronological data, the evolution of palaeo-dam and the causes of terrace formation might need to be further studied, and the roles of tectonic activity, climate, and blocking-

outbursting events on the formation of terraces should be considered.

To explore the above unsolved problems, we investigated the geomorphological and sedimentological characteristics of the Tuanjie and Taiping Terraces, with two independent dating methods by optically stimulated luminescence (OSL) and radiocarbon. The purposes of this paper are: (1) to clarify the deposition ages and sedimentary characteristics of Taiping and Tuanjie terraces; (2) to

reveal the blockage and outburst of the palaeo-dam; (3) to explore the influences of tectonics, climate, and geological disasters (blocking and damming) on the formation of terraces.

## 2 Study area

Diexi area lies in the upper Minjiang River. The upper Minjiang River belongs to the northeast

margin of the Tethys Himalayan domain and the Barkam formation zone, on the eastern margin of the Bayan Har Block (Fig. 1a). The Minjiang valleys, narrow at higher altitudes, and gradually widen downstream. Their width varies from 60 to 300 m (Yang, 2005; Jiang et al., 2016; Ma, 2017; Zhang, 2019), and the steep hillslopes on both sides of the river valley have a slope of 30-35° (Zhang et al., 2011; Guo, 2018), with a depth of 800 to 3000 m. There are many outburst sediments deposited downstream

of Diexi, such as Xiaoguanzi, Shuigouzi, and Manaoding (Fig. 1a).

Diexi palaeo-dammed lake (31 º26′-33 º16′ N; 102 º59′-104 º14′ E) situated on the bend of the V-shape Minjiang valley, which in turn lies in the well-known "north–south earthquake tectonic zone"



(Tang et al., 1983; Huang et al., 2003; Yang, 2005; Deng et al., 2013). The palaeo-landslide dam that formed the Diexi palaeo-lake is located on the left bank of the Minjiang River, from the Jiaochang to the

Diexi ancient town (Fig. 1b). The highest elevation is about 3390 m, and the main slide direction is about SW18°. The length and width of the palaeo-landslide are respectively about 3500 m and 3000 m, with a volume of the accumulation reaching $1.4\sim2.0\times10^9$ m³ (Zhong et al., 2021).

Diexi is being forged in the collision of the Indian and Eurasian plates (Fig. 1b), in the eastern Tibetan Plateau. As the Tibetan Plateau and its surrounding areas have been affected by strong and

frequent earthquakes during the late Quaternary (Yang et al., 1982; Chen and Lin, 1993; Li and Fang, 1998; Shi et al., 1999; Hou et al., 2001; Lu et al., 2004), Lake Diexi is affected by active and accelerated tectonics. The area features visible strata from various periods: Devonian, Carboniferous, Permian, Triassic, and Quaternary (An et al., 2008; Zhang et al., 2011; Ma, 2017; Zhong, 2017). The Songpinggou River flows eastward as the tributary of the Minjiang River and merges into the Minjiang River in Lake

Diexi. It has a typical alpine landform erosion with an elevation of 1868-4800 m. Large amounts of Triassic sediments are deposited along the Songpinggou river bed.

The climate of the entire region is monsoonal, as it is interested in the Plateau Monsoon, the Westerlies, and the East Asian Monsoon. The Diexi Valley, due to atmospheric circulation and the mountainous character, shows an arid and semi-arid climate (Shi, 2020). With the strong effect of the

prevailing winds, the annual cumulative evaporation can reach 1000-1800 mm (Yang, 2005), and the average temperature and precipitation are 13.4°C and 500-600 mm, respectively. In terms of ecological pattern, the vegetation in the study area shows a visible vertical zonation, largely composed of mountain coniferous forest, alpine meadow, and low shrubs. The Songpinggou areas are scattered with forests of mountain pinus tabulaeformis, Sichuan-Yunnan alpine oak evergreen shrubs, and forests of deciduous

species such as poplar and birch (Shi, 2020).

The seven staircases of Tuanjie Terrace (32º2.7′ N, 103º40.1′ E) are located in Tuanjie village, on the right bank of the Minjiang River, at the mouth of the Songpinggou tributary (Fig. 1c). The three staircases of Taiping Terrace (32º12′13.1″ N, 103º45′53.7″ E) are in the Taiping village, at the mouth of Luobogou gully, which is 12 km upstream of the Tuanjie Terrace (Fig. 1d) (Fan et al., 2021; Wang et al.,

2005b). The course of the river from Taiping to Manaoding is a deep canyon (Duan, 2002), albeit the Taiping and Tuanjie area is a wide valley landform.



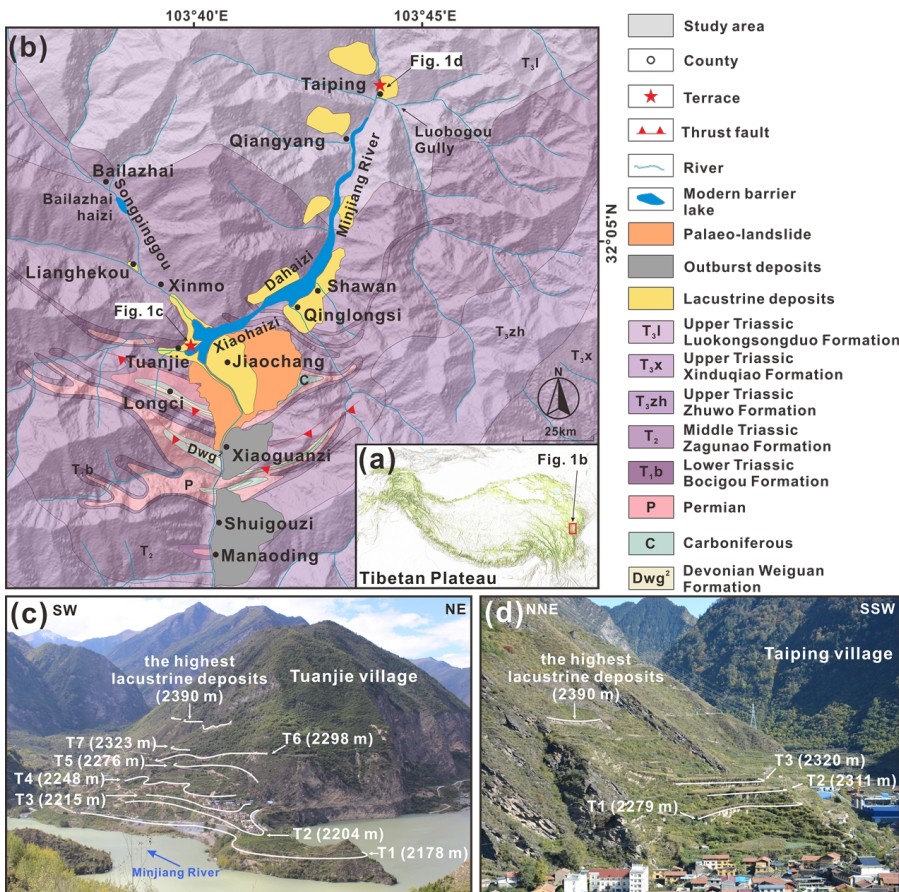

**Figure 1. Location of the study area. (a) Overview of the Diexi area at the eastern margin of the Tibetan**
**Plateau. (b) Geological characteristics of Diexi area and legends (maps modified from Guo, 2018; Wang et al.,**
**2020; Zhong et al., 2021). (c) The topography of the Tuanjie Terrace. (d) The topography of the Taiping**
**Terrace. The elevation of each terrace level is shown in (c) and (d).**

## 3 Materials and methods

### 3.1 Geomorphic and sedimentary description

From October to November 2018, field surveys were carried out in the Diexi. These terraces are
named in order of Terrace 1 (T1, lowest position with oldest age) to Terrace 7 (T7, highest terrace level
and youngest). The sedimentary structure, geometric shape, sorting, roundness, and the direction of
gravels are described. The lithofacies of Diexi palaeo-dammed lake were analyzed by the classification
method of sedimentary facies (Miall, 2000) and by previous research in the Diexi area (Yang, 2005;



Yang et al., 2008) (Table. 1).

**Table. 1 Lithofacies of Diexi area, eastern Tibet Plateau. Adapted from Miall (2000), Yang (2005) and Yang et al. (2008).**

| Lithofacies code | Lithofacies | Sedimentary structures | Interpretation |
|---|---|---|---|
| Ps | Paleosol | Pedogenic features, roots | Pedogenesis |
| Ls | Sandy loess | Massive texture | Eolian deposits |
| Gmm | Matrix-supported, massive gravel | Weak grading | Plastic debris flow (high-strength, viscous) |
| Gh | Clast-supported, crudely bedded gravel | Horizontal bedding, imbrication | Longitudinal bedforms, lag deposits, sieve deposits |
| Gci | Clast-supported gravel | Inverse grading | Clast-rich debris flow (high strength), or pseudoplastic debris flow (low strength) |
| Gcm | Clast-supported, massive gravel | - | Pseudoplastic debris flow (inertial bedload, turbulent flow) |
| Fm | Mud | snail shells | Overbank, abandoned channel, or drape deposits |
| Fl | silty clay | parallel bedding, wave bedding | Lacustrine deposits |


### 3.2 Chronology

To obtain a reliable chronostratigraphic framework, two independent dating methods were used: optically stimulated luminescence (OSL) and radiocarbon. A total of twenty-two samples were taken from the Tuanjie and Taiping terraces. Of these, nineteen have been dedicated to OSL dating and the

remaining three to radiocarbon (Fig. 2).

### 3.2.1 OSL

Nineteen OSL samples were collected from lacustrine deposits, gravel units, loess, and paleosol (Fig. 2 and 3). In Tuanjie Terrace, twelve samples were collected from the lacustrine deposits and the

paleosol of T1 to T5 and T7 terraces, and two samples from the gravel units of T2 and T5 (Fig. 2 and 3). In Taiping Terrace, four samples were taken from the lacustrine deposits at T1 to T3 terraces and the highest deposits, another one was taken from the paleosol unit at T3 terrace (Fig. 2 and 3). To ensure that human activities and weathering did not disturb the samples, we scraped the surface sediments, and





pushed the stainless steel tubes with a hammer to collect shielded sediments. Both ends of the tube have

been sealed with black opaque paper.

Samples were processed and measured at the Luminescence Geochronology Laboratory at the Institute of Earth Environment, Chinese Academy of Sciences. The pre-treatment was carried out to obtain fine-grained (4-11 μm) and coarse-grained (90-150 μm) mixed mineral fractions from the samples (Aitken, 1985, 1998; Lu et al., 1988). These fine-grained mixed mineral fractions were then immersed

in fluorosilicic acid for 3 days to extract the fine-grained quartz by dissolving the feldspar; finally, the fine-grained quartz was precipitated on a 9.7 mm diameter stainless steel sheet using acetone for experimental use (Wang et al., 2005c). The laboratory uses ultrasonic waves to continuously eliminate the binding phenomenon in the process of extracting fine grains of quartz from mixed minerals and separating the fine particles of quartz that met the requirements of luminescence dating (Wang et al.,

2005d). For coarse particles, organic matter and carbonates were removed with hydrogen peroxide and hydrochloric acid, etched with hydrofluoric acid for 60 mins, and then washed with distilled water. The magnetic minerals were removed by drying at low temperatures, and the remaining pure quartz sample was fixed on a stainless steel test piece with silica gel (Zhu et al., 2011).

Luminescence signals were measured on a Daybreak 2200 automated measurement system, and

detected by an EM IQA9235 photomultiplier tube (PMT). The excitation sources were blue light at (470 ± 5) nm and infrared light at (880 ± 60) nm with an excitation power of about 45 mW cm$^{-2}$, and the excitation temperature was 125°C. The excitation time for the measurement was 50 s. Two U-340 filters (3 mm thick) were attached between the excitation source and the PMT to isolate the excitation source signal and other spurious signals. The equivalent dose was calculated as the integrated value of the first

5 s of the decay curve minus the integrated value of the last 5 s as the background (Wang et al., 2005c). Water content and cosmogenic nuclides also test in this laboratory.

### 3.2.2 Radiocarbon

Three samples are obtained for radiocarbon analysis, including two samples of the highest lacustrine

deposits in the Tuanjie and Taiping Terraces, and one is from the loess that overlain the T5 terrace in Tuanjie (Fig. 2 and 3). To avoid the influence of weathering, the surface sediments were removed.

All the samples were tested for organic matter, and analyzed by using the NEC accelerator mass



spectrometer and Thermo infra-red mass spectrometer at the Beta Analytic Radiocarbon Dating

Laboratory. Samples were pre-treated following their protocols. The ages were converted into calendar

years by using the IntCal 20 calibration curve (Reimer et al., 2020).

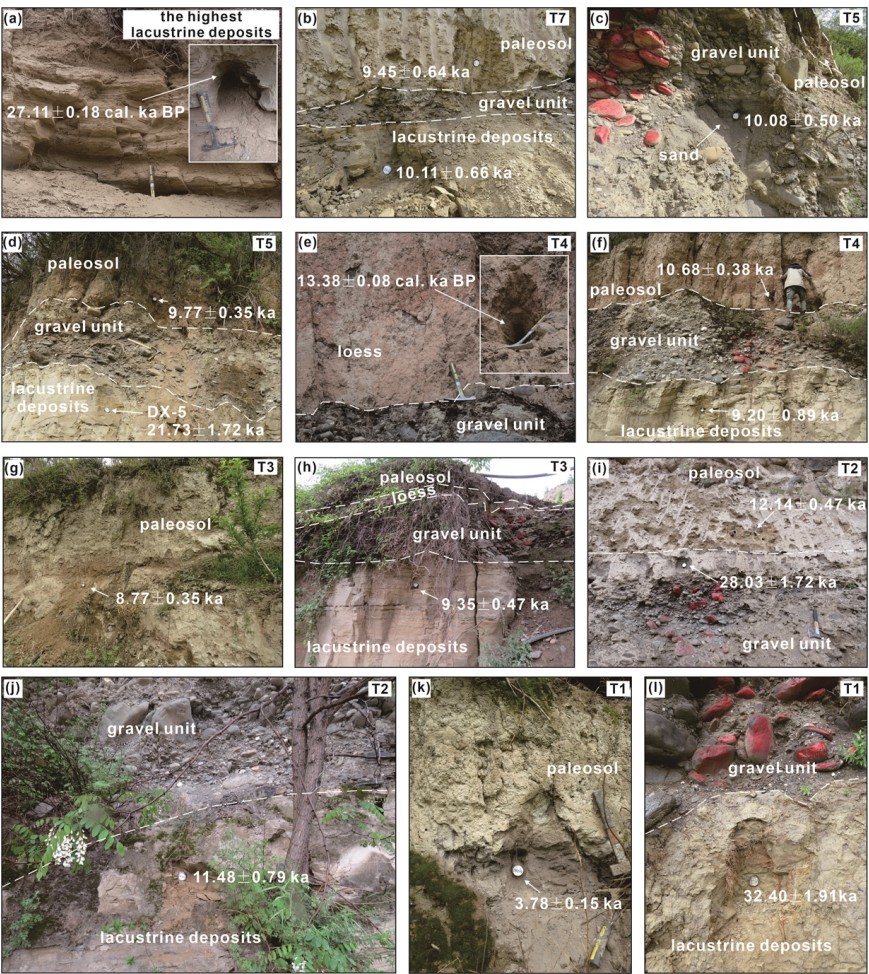

**Figure 2. OSL and radiocarbon samples from Tuanjie Terrace. (a) Radiocarbon sample of the highest lacustrine deposits. (b) OSL samples of the lacustrine deposits and paleosol in the T7. (c) OSL sample of the gravel unit in the T5. (d) OSL samples of the lacustrine deposits and paleosol in the T5. (e) Radiocarbon sample of loess in the T4. (f) OSL samples of the lacustrine deposits and paleosol in the T4. (g) OSL sample of the paleosol in the T3. (h) OSL sample of the lacustrine deposit in the T5. (i) OSL samples of the gravel unit and paleosol in the T2. (j) OSL sample of the lacustrine deposits in the T2. (k) OSL sample of the paleosol in the T1. (l) OSL sample of the lacustrine deposits in the T1. The white dashed line marks the boundary between units (same as below).**



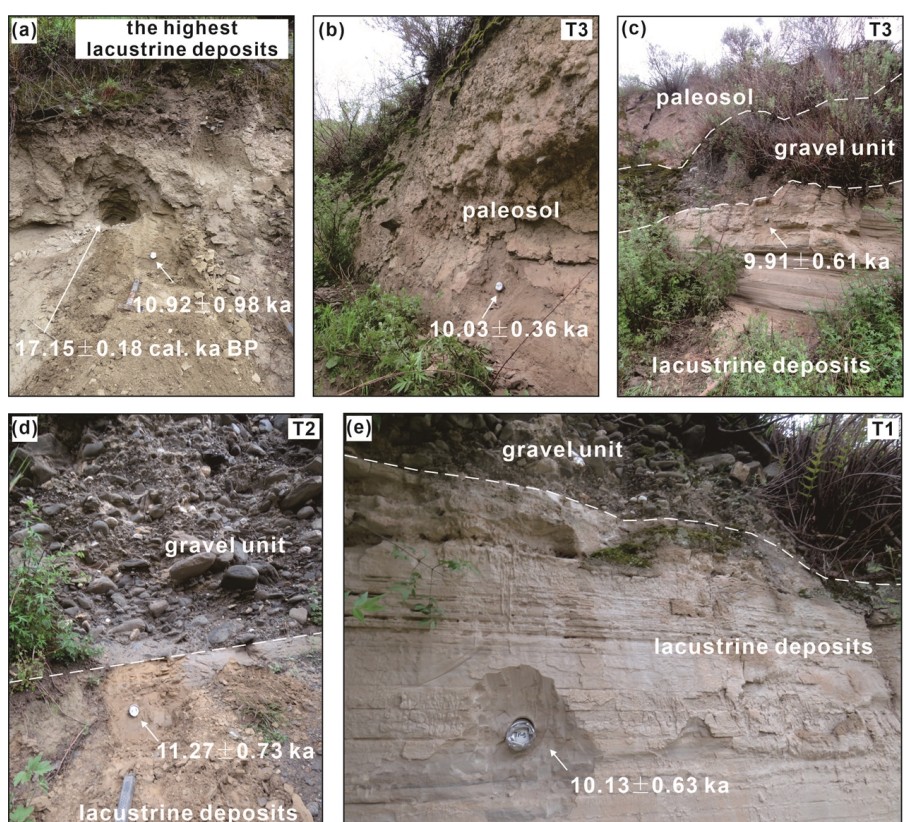

**Figure 3. OSL and radiocarbon samples were taken from Taiping Terrace. (a) The paired OSL and radiocarbon samples were collected from the highest lacustrine deposits. (b) OSL sample of paleosol in the T3. (c) OSL sample of the lacustrine deposits in the T3. (d) OSL sample of the lacustrine deposits in the T2; (e) OSL sample of the lacustrine deposits in the T1.**

## 4 Results

### 4.1 Terraces distribution and sequences

#### 4.1.1 Terraces distribution

Tunajie Terrace has seven staircases, Taiping Terrace has three staircases, and all these terraces are overlain on the lacustrine deposits (Fig. 4). The thickness of lacustrine deposits in Tuanjie Terrace is >200 m, and the lateral lengths of the seven terraces range from 150 to 1000 m (Fig. 4), Terrace T1 has the largest extension, which extends toward the center of the Diexi Lake (Fig. 1c). Taiping terraces developed on a high mountain with a slope of 40°-60°, influenced by landslides and croplands. The horizontal



extensions of T1, T2, and T3 are equal to 520 m, 380 m, and 190 m, respectively.

Terrace elevations are obtained by using the Light Detection And Ranging (LiDAR) with 0.5 m accuracy and the Advanced Spaceborne Thermal Emission and Reflection Radiometer Global Digital Elevation Model (ASTER GDEM) with 30 m accuracy (Fan et al., 2021). Data were imported in ArcGIS 205 10.3, then field investigations on two Terraces determined their altimetric level (Table. 2), textures, and forming ages (Fig. 4). The elevation data reported in Fig. 1c and 1d shows the elevations of all the terrace surfaces.

**Table. 2 Elevation of the Tuanjie and Taiping terraces.**

| Tuanjie Terrace | Elevation (m) | Taiping Terrace | Elevation (m) |
|---|---|---|---|
| Highest | 2390 | Highest | 2390 |
| T7 | 2323 | T3 | 2320 |
| T6 | 2298 | T2 | 2311 |
| T5 | 2276 | T1 | 2279 |
| T4 | 2248 | - | - |
| T3 | 2215 | - | - |
| T2 | 2204 | - | - |
| T1 | 2178 | - | - |


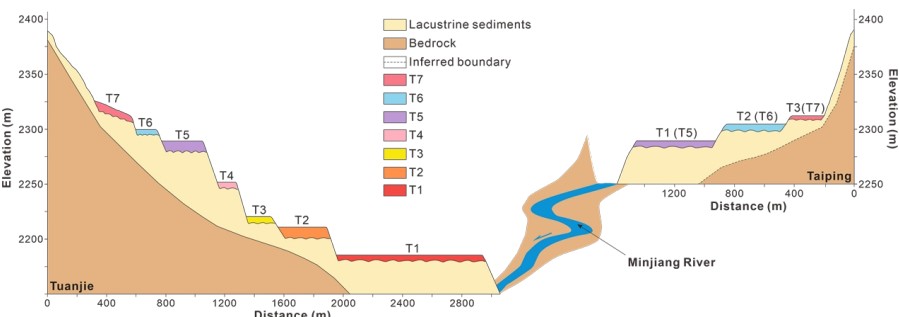

**Figure 4. Correlation between the Tuanjie and Taiping terraces. Elevation data showed that Taiping T1 to T3 corresponded to Tuanjie T5 to T7, respectively.**

**4.1.2 Terraces lithostratigraphy**

We summarized the lithology, texture, and sedimentary structures of the Tuanjie and Taiping terraces.



**Tuanjie Terrace**

The lithostratigraphy of Tuanjie terraces from bottom to top is (Table. 1 and Fig. 5a): (1) Silt clay (*Fl*) with strong weathering, horizontal bedding, and wave bedding denote the presence of lacustrine deposits. (2) Gravel unit (*Gh, Gci, Gmm*) is fluvial deposits, featuring an unconformity with the underlayer. The orientation of gravels is mostly parallel to the Minjiang River, suggesting the source of these gravels is the Minjiang River. Gravel units in Tuanjie T1, T3, T4, and T5 (*Gh*) are generally poorly

sorted and well-rounded, with a diameter of 2-30 cm, indicative of longitudinal bedforms, lag deposits, and sieve deposits (Fig. 5a). Gravels in Tuanjie T2 (*Gci*) are 2-25 cm in diameter with inverse grading, the gravel with 35 cm diameter is rare, poorly sorted and sub-circular to round gravels without direction. These features support this gravel unit clast-rich debris flow with high strength or pseudoplastic debris flow with low strength. Gravels in Tuanjie T6 (*Gmm*) have graded bedding with well sorting and

rounding, interpreting that these gravels deposited by the plastic debris flow with high strength. (3) Loess (*Ls*), some of which are brick-red loess. Angular phyllites are appears in T3. (4) Paleosol (*Ps*) capping the top of each terrace, roots are abundant in this unit (Fig. 5a). The lacustrine deposits with a thickness of 30 m also appear above T7, with undulating bedding and severe denudation, reaching up to 2390 m at the highest point (Fig. 5a).

However, the strata of each terrace are different, terraces T1, T2, T3, T4, and T6 are covered by gravels, loess, and paleosol units, while T5 and T7 lack the loess unit (Fig. 5a). Terraces T4 to T7 have been deformed and collapsed in varying degrees due to cultivation and excavation, human activity and disaster events may be the cause of deformation.

**Taiping Terrace**

     The set of three terraces in Taiping also has the base of lacustrine deposits (Fan et al., 2021), Terraces T1 and T2 have similar sedimentary sequences with the T5 and T6 of the Tuanjie Terrace (Fig. 5). Taiping T1 is covered by gravels and paleosol, T2 is covered with gravels, loess and paleosol (Fig. 5b). Terrace T3 has different features: gravel unit (*Gcm*) is overlain by two sequences of mud (*Fm*)-

phyllite clasts (*Gh, Gci*) layers (Fig. 5b). Snail shells can be seen in the mud, and the two clast layers are composed of neatly arranged phyllites (Fig. 5b).

     Three gravel units of Taiping terraces have the direction along the Luobogou Gully or without



direction, this means that these deposits originated from the gully with a high-energy event. Gravels in

T1 of Taiping (*Gcm*) are featured with poorly sorted and subrounded gravels with a diameter of 5-10 cm,

implying that this is a pseudoplastic debris flow (Fig. 5). The gravel units in Taiping T2 and T3 (*Gcm*)

have many broken phyllites, interpreted as pseudoplastic debris flows. Loess units in Taiping T2 and T3

(*Ls*) are mixed with 2-5 cm diameter of angular phyllites, this feature indicates that there have been high

energy environments to mix the loess with phyllite clasts. The two sequences of mud-phyllite clasts

layers in Taiping T3 indicate that there have been two blocking events, as snail shells in the mud mark

the occurrence of an overbank, an abandoned channel, or drape deposits (Fig. 5).

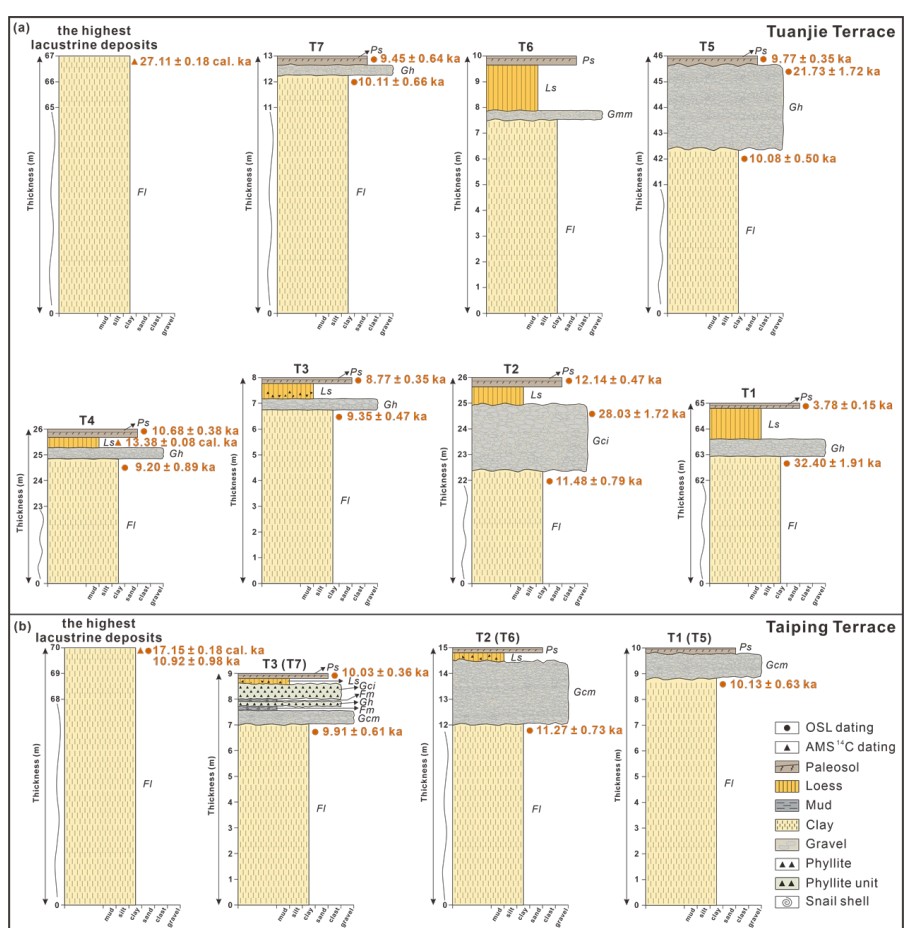

**Figure 5. Sedimentary deposits, lithofacies, and chronologies of the Tuanjie and Taiping terraces. (a) T1, T2, T3, T4, T5, T6, T7, and the highest lacustrine deposits of Tuanjie terraces, respectively. Terraces T1 to T4**

**and T6 are composed, from bottom to top, of lacustrine deposits, gravels, loess, and paleosol. The Loess of T3**



**contains phyllites. The sedimentary sequence of T5 and T7 are composed of lacustrine deposits, gravels, and paleosol. (b) T1, T2, T3, and the highest lacustrine deposits of Taiping terraces. Terrace T1 is composed of lacustrine deposits, gravels, and paleosol, from bottom to top. The sequence of T2 is lacustrine deposits, gravels, loess, and paleosol. In T3, there are two sets of mud-phyllite clasts layers deposited between the gravel**

**unit and loess. Loess in T2 and T3 mixed with phyllites. Terraces T1 to T3 in Taiping correspond to terraces T5 to T7 in Tuanjie. OSL and radiocarbon dating results are denoted. Lithofacies coding as Table. 1.**

### 4.2 OSL ages

A total of nineteen OSL samples were collected, divided into fourteen samples from Tuanjie and

five samples from Taiping (Table. 3). Radionuclides ($^{40}$K, $^{232}$Th, $^{238}$U) have been measured for all the samples, and the equivalent doses (De) were calculated.

Optional Stimulated Luminescence (OSL) dating of lacustrine deposits in Tuanjie terraces yielded ages of 32.40±1.91 ka for the T1, 11.48±0.79 ka for the T2, 9.35±0.47 ka for the T3, 9.20±0.89 ka for the T4, 10.11±0.66 ka for the T5 and 10.08±0.5 ka for the T7. Consequently, T1 was deposited in the

late Pleistocene; T2, T5, and T7 were deposited at the beginning of the Holocene; and T3 and T4 were deposited around 9.00 ka. Dating results of gravels from the T2 and T5 show that these were deposited in the late Pleistocene, and the ages are 28.03±1.72 ka and 21.73±1.72 ka, respectively. The ages of paleosol capped on the top of each terrace are different, but all deposited during the Holocene. Paleosol of T2 and T4 were deposited with the older ages of 12.14±0.47 ka and 10.68±0.38 ka, respectively. And

the paleosol unit of T3, T5, and T7 are deposited at 8.77±0.35 ka, 9.77±0.35 ka, and 9.45±0.64 ka, respectively. T1 has the youngest paleosol unit with an age of 3.78±0.15 ka. Terraces grow younger amid the increase in elevation.

The OSL ages of lacustrine deposits of Taiping T1 to T3 are 10.13±0.63 ka, 11.27±0.73 ka, and 9.91±0.61 ka, respectively. The highest lacustrine deposits showed an age of 10.92±0.98 ka. All the

terraces were deposited during the Holocene. Loess in T3 is 10.03±0.36 ka which is similar to the ages of lacustrine deposits of T1 and T2.





**Table. 3 OSL ages, radioisotope, water content, and dose rate of OSL samples at the Tuanjie and Taiping Terraces.**

| Location | Deposit level | Sample | Lab code | Facies | Longitude and latitude | Elevation (m) | Depth (m) | Grain size (μm) | Water content (%) | K (%) | Th (ppm) | U (ppm) | Dose rate (Gy/ka) | Equivalent dose (Gy) | OSL age (ka) |
|---|---|---|---|---|---|---|---|---|---|---|---|---|---|---|---|
| Taiping | - | TP19-1 | IEE5554 | lacustrine | 32°7'37.07"N, 103°44'13.99"E | 2342.95 | 1.90 | 4-11 | 20±5 | 1.98±0.03 | 12.85±0.37 | 4.82±0.14 | 4.27±0.14 | 46.59±3.88 | 10.92±0.98 |
| | T3 | TP19-2 | IEE5555 | paleosol | 32°7'33.65"N, 103°44'11.93"E | 2279.14 | 3.50 | 4-11 | 10±5 | 2.01±0.02 | 14.75±0.20 | 2.92±0.05 | 4.28±0.15 | 42.95±0.23 | 10.03±0.36 |
| | | TP19-3 | IEE5556 | lacustrine | | | 4.20 | 4-11 | 20±5 | 2.20±0.04 | 14.23±0.27 | 3.62±0.55 | 4.17±0.16 | 41.33±1.97 | 9.91±0.61 |
| | T2 | TP19-4 | IEE5557 | lacustrine | 32°7'32.31"N, 103°44'10.68"E | 2219.57 | 3.60 | 4-11 | 20±5 | 1.90±0.01 | 12.59±0.40 | 3.29±0.10 | 3.71±0.12 | 41.81±2.35 | 11.27±0.73 |
| | T1 | TP19-5 | IEE5558 | lacustrine | 32°7'32.81"N, 103°44'10.53"E | 2177.27 | 1.00 | 4-11 | 20±5 | 2.17±0.02 | 12.74±0.19 | 3.31±0.07 | 4.02±0.13 | 10.74±2.18 | 10.13±0.63 |
| Tuanjie | T7 | DX19-1 | IEE5540 | paleosol | 32°2'42.14"N, 103°39'45.31"E | 2315.45 | 2.30 | 4-11 | 10±5 | 2.16±0.07 | 13.86±0.28 | 3.48±0.04 | 4.54±0.17 | 42.92±2.45 | 9.45±0.64 |
| | | DX19-2 | IEE5541 | lacustrine | | | 2.90 | 4-11 | 20±5 | 2.40±0.05 | 14.00±0.20 | 3.41±0.05 | 4.29±0.14 | 43.37±2.46 | 10.11±0.66 |
| | | DX19-4 | IEE5543 | paleosol | 32°2'42.44"N, 103°39'48.35"E | 2265.88 | 1.30 | 4-11 | 10±5 | 2.03±0.02 | 13.49±0.21 | 2.93±0.07 | 4.25±0.15 | 41.47±0.17 | 9.77±0.35 |
| | T5 | DX19-3 | IEE5542 | fluvial | 32°2'45.99"N, 103°39'55.24"E | 2264.93 | 2.60 | 90-150 | 10±5 | 1.53±0.04 | 10.25±0.17 | 2.28±0.05 | 2.70±0.11 | 58.71±4.02 | 21.73±1.72 |
| | | DX19-5 | IEE5544 | lacustrine | 32°2'42.44"N, 103°39'48.35"E | 2265.88 | 2.80 | 4-11 | 20±5 | 2.16±0.05 | 13.34±0.13 | 3.14±0.05 | 3.96±0.13 | 39.94±1.45 | 10.08±0.50 |
| | | DX19-6 | IEE5545 | paleosol | | | 2.20 | 4-11 | 10±5 | 2.00±0.01 | 14.35±0.10 | 2.85±0.03 | 4.24±0.15 | 45.33±0.16 | 10.68±0.38 |
| | T4 | DX19-7 | IEE5546 | lacustrine | 32°2'39.76"N, 103°39'52.59"E | 2228.79 | 5.00 | 4-11 | 20±5 | 2.45±0.04 | 14.13±0.36 | 3.57±0.06 | 4.34±0.15 | 39.95±3.61 | 9.20±0.89 |
| | T3 | DX19-8 | IEE5547 | paleosol | 32°2'40.00"N, 103°39'54.67"E | 2192.44 | 2.20 | 4-11 | 10±5 | 1.96±0.07 | 12.78±0.19 | 2.99±0.29 | 4.11±0.16 | 36.05±0.15 | 8.77±0.35 |
| | | DX19-9 | IEE5548 | lacustrine | | | 2.10 | 4-11 | 20±5 | 2.48±0.02 | 13.54±0.21 | 3.12±0.16 | 4.26±0.14 | 39.80±1.50 | 9.35±0.47 |
| | T2 | DX19-10 | IEE5549 | paleosol | | | 5.00 | 4-11 | 10±5 | 2.41±0.06 | 13.97±0.23 | 3.40±0.05 | 4.70±0.18 | 57.06±0.53 | 12.14±0.47 |
| | | DX19-11 | IEE5550 | fluvial | 32°2'46.42"N, 103°39'59.89"E | 2180.47 | 5.50 | 90-150 | 10±5 | 1.78±0.04 | 14.74±0.12 | 3.37±0.04 | 3.38±0.13 | 94.60±4.51 | 28.03±1.72 |
| | T1 | DX19-12 | IEE5551 | lacustrine | 32°2'42.34"N, 103°40'07.75"E | 2193.80 | 4.50 | 4-11 | 20±5 | 2.26±0.07 | 13.76±0.16 | 3.35±0.04 | 4.10±0.14 | 47.07±2.81 | 11.48±0.79 |
| | | DX19-14 | IEE5553 | paleosol | 32°2'41.16"N, 103°40'10.51"E | 2148.88 | 0.60 | 4-11 | 10±5 | 1.52±0.07 | 8.69±0.29 | 2.38±0.07 | 3.20±0.13 | 12.09±0.04 | 3.78±0.15 |
| | | DX19-13 | IEE5552 | lacustrine | 32°2'42.73"N, 103°40'13.06"E | 2150.60 | 2.50 | 4-11 | 20±5 | 2.16±0.05 | 11.81±0.10 | 2.89±0.03 | 3.77±0.13 | 122.24±5.93 | 32.40±1.91 |

* Terraces are not completely flat, so the elevation data of some samples deviate from the elevation of the terrace.



### 4.3 AMS ¹⁴C ages

The highest lacustrine deposits of the Tuanjie and Taiping Terraces were deposited at 27.11±0.18 cal. ka BP and 17.15±0.18 cal. ka BP, respectively. The loess of T4 at Tuanjie was developed at 13.38±0.08 cal. ka BP (Table 4.).

**Table. 4 Radiocarbon results for the Tuanjie and Taiping Terraces.**

| Samples | Lab code | Material | Elevation (m) | δ¹³C (‰) | Radiocarbon age (a BP) | Calibration age (cal. ka BP) |
|---------|----------|----------|---------------|-----------|------------------------|-------------------------------|
| TP-max | Beta-520926 | bulk sediment | 2342.95 | -19.1 | 14050±50 | 17.15±0.18 |
| TJ-max | Beta-520925 | bulk sediment | 2390.00 | -19.2 | 22740±90 | 27.11±0.18 |
| TJ-T4-HT | Beta-520924 | bulk sediment | 2280.00 | -21.6 | 11490±40 | 13.38±0.08 |


## 5 Discussion

### 5.1 Reliability of dating results

In our study, a total of three AMS ¹⁴C samples and nineteen OSL samples of lacustrine deposits were collected from the Tuanjie and Taiping Terraces (Tables. 3, 4). Lacustrine deposits extracted from

Tuanjie T1 were deposited at 32.40±1.91 ka (DX19-13, Fig. 5a). This excessively old age indicates that the sample is affected by poor bleaching of the sediments, which may have been caused by particle size or transport conditions (Long et al., 2012; Zhao et al., 2014). Nevertheless, the DX19-13 sample is reliable because this age is contemporary to the bottom age of the ZK2 core (35.13±0.29 cal. ka BP, Wang et al., 2012) and the top and bottom ages of the Tuanjie section (35.78±0.37~30.66±0.03 cal. ka

BP, Zhang et al., 2009). Compared with all the ages in the Tuanjie Terraces, gravel units have the older ages, they are 28.03±1.72 ka (Tuanjie T2) and 21.73±1.72 ka (Tuanjie T5) (see Fig. 5). Although fluvial sediments are susceptible to poor bleaching, the ages of gravels is similar to that of convolution structuress in Haizipo (27.24±0.41 cal. ka BP, 27.74±0.47 cal. ka BP; Wang et al., 2012). Therefore, gravel units in the Tuanjie T2, T5, and convolution structures in Haizipo may have been formed at the

same time, suggesting that the gravel ages are reliable.

The highest lacustrine deposits in Taiping were subject to paired radiocarbon and OSL dating, with



ages being 17.15±0.18 cal. ka BP and 10.92±0.98 ka, respectively (Fig. 5). The discrepancy shows that the radiocarbon age appears ~6 ka older than the OSL age. Considering that the radiocarbon ages are sampled from the bulk sediments, the radiocarbon ages might be overestimated due to the old carbon

effect. The reservoir effect affects the overestimation of age because of: (1) The $^{14}$C specific activity in water is lower than that in the atmosphere (Deevey et al., 1954). (2) Subaqueous landslides, slumps, or other disturbances might have mixed older sediments with younger ones (Counts et al., 2015; Shi, 2020). (3) The re-deposition of older organic components affects the biological indicators of sediments toward a pre-dating bias (Kaplan et al., 2017; Krivonogov et al., 2016).

Besides, previous studies show the terrace ages of the lacustrine deposits along the Diexi area are mainly between 35.78 and 10.63 ka (Table. S1). Our dating results of lacustrine deposits lie within this range, supporting our data are reliable.

**5.2 Evolution of terraces in the upper Minjiang River**

**5.2.1 Terraces along the upper Minjiang River**

Along the upper Minjiang River, there are at least twenty-two terraces. Eleven terraces are distributed upstream of the Diexi area (from Zhangla to Gonggaling), four sites near the Diexi area (Taiping, Shawan, Jiaochang, and Tuanjie areas), and seven sites are developed downstream (from Maoxian-Wenchuan). There are 124 terrace dating samples from these sites (Table. S1), including thirty-

two samples from the upstream, eighty-two samples from the study area, and nine samples from the downstream. These terrace ages of the upstream reveal that the formation and evolution of terraces in the upper Minjiang River began in 830.00 ka (the early Pleistocene) (Table. S1), and mainly formed between 47 to 2 ka (Fig. 6). In the Diexi area, the ages of these terraces are distributed at 550-50 ka (Table. S1), and mainly are 32-2 ka (Fig. 6). The downstream terraces were deposited at 400-50 ka (Table. S1), mostly

were deposited at 40-20 ka (Fig. 6). In summary, terraces ages of the upper Minjiang River range from 830.00 to 0.91 ka, mostly were formed between 39.90 and 6.12 ka. Compared with the upstream and downstream, more terraces developed in the Diexi area, and these terraces were mainly formed during 30~0 ka.

Terraces of the area from Zhangla basin to the source of the Minjiang River are considered as a

result of tectonic uplift (Yang et al., 2003; Yang, 2005; Yang et al., 2011; Yang et al., 2008; Chen and



Li, 2014; Zhu, 2014). In the Diexi area, the formation of the Tuanjie and Taiping Terraces was impacted

by the evolution of the palaeo-dam (Duan et al., 2002; Wang et al., 2005b; Wang, 2009; Zhu, 2014). The

formation of terraces in Maoxian-Wenchuan is similar to that of Diexi terraces, they are also the result

of the outburst of a palaeo-dammed lake (Zhu, 2014). It can be seen that the terrace formation mechanism

in the study area and downstream is different from that in the upstream. However, previous studies have

not provided enough evidence to support this perspective, and we will publish more evidence in the

following sections.

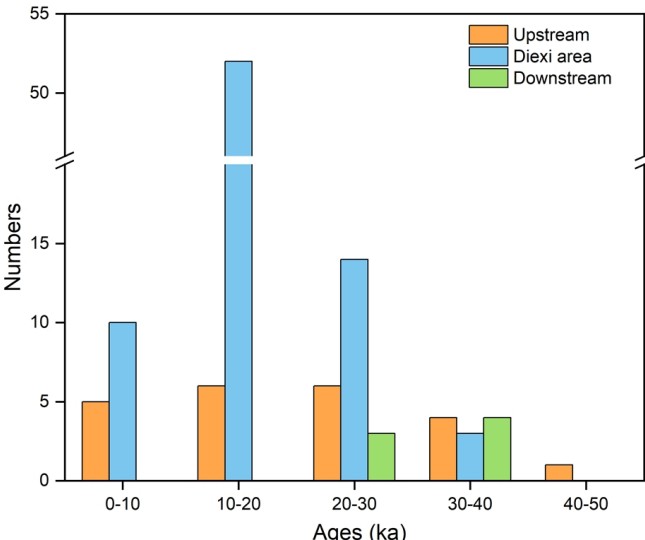

**Figure 6. Frequency distribution histogram of terrace ages formed since 50 ka in the upper reaches of the**
**Minjiang River, these ages are distributed between 46.40 to 2.81 ka. The terraces ages of the upstream area**
**are distributed between 46.40 ka to 2.81 ka. The terraces ages of the Diexi area range from 35.78 and 3.70 ka.**
**The ages of the terraces in the downstream are distributed in 39.90 to 20.70 ka. Terraces developed in the**
**Diexi area were mainly formed during 30~8 ka.**


### 5.2.2 Correlation of the Tuanjie and Taiping Terraces

The highest lacustrine deposits in Tuanjie and Taiping have equal elevation (2390 m) suggests that

Taiping Terrace and Tuanjie Terrace are somehow related.

Other evidence comes from the combination of elevation and stratigraphy, thus the T1 to T3 terraces

of Taiping correspond to the T5 to T7 terraces of Tuanjie (Fig. 4). Terrace T5 (Tuanjie) and T1 (Taiping)



have the same sedimentary sequences from the bottom to the top, including clays (*Fl*), gravel unit (*Gh* in Tuanjie, *Gcm* in Taiping) and paleosol (*Ps*) (Fig. 5a and 5b). Paleosol (*Ps*) in T5 (Tuanjie) and T1 (Taiping) are 0.4 m and 0.2 m thick, respectively. Both T6 (Tuanjie) and T2 (Taiping) have the sequences of clays (*Fl*), gravel unit (*Gmm* in Tuanjie, *Gcm* in Taiping), loess (*Ls*), and paleosol (*Ps*). The

sedimentary successions of Terrace T7 (Tuanjie) and T3 (Taiping) both are clays (*Fl*), gravel unit (*Gh* in Tuanjie, while *Gcm* in Taiping), and paleosol (*Ps*). T3 of Taiping has two sets of mud-phyllite clasts (*Fm-Gh* and *Fm-Gci*) overlaying the gravel unit (*Gcm*), and loess (*Ls*) contains phyllites. These different lithofacies and sequences are caused by regional geomorphic environments.

The chronological data confirmed that the Taiping T1 to T3 are matched with the Tuanjie T5 to T7

(Fig. 5). Ages of the lacustrine deposits in T5 (Tuanjie) and T1 (Taiping) are 10.08±0.50 ka and 10.13±0.63 ka. For Terrace T7 (Tuanjie) and T3 (Taiping), the dating results of lacustrine deposits are 10.11±0.66 ka and 9.91±0.61 ka. Also, the deposited ages of the paleosol in T7 (Tuanjie) and T3 (Taiping) are 9.45±0.64 ka and 10.03±0.36 ka.

**5.3 Formation and outburst of palaeo-dam**

The triangle formed by Tuanjie and the localities of Jiaochang and Xiaohaizi lies around the center of the ancient dammed lake. Outburst sediments are present downstream around the location of Xiaoguanzi-Manaoding (Fig. 1b). Combined with the lithofacies and chronological framework in the Tuanjie terraces, the palaeo-dam has experienced multiple events of damming and dam-breaking.

The palaeo-landslide dam blocked the river before 32.40±1.91 ka, as Wang et al. (2012) and Wang et al. (2017) support: (1) The bottom lacustrine deposits of the palaeo-dammed lake was deposited at 35.13±0.29 cal. ka BP. (2) The boundary of the palaeo-dammed lake and palaeo-dam in Xiaoguanzi is deposited at 34.87±0.76 and 35.54±0.83 cal. ka BP. (3) Accumulation deposits of palaeo-dam in Manaoding are deposited at 34.54±0.16 cal. ka BP. (4) The strong activity of the Maoxian-Wenchuan

fault happened at 45/40-20 ka ago.

After the first dammed lake phase, the first outburst occurred at 27.11±0.18 ka, as the outburst sediments in the downstream deposited at 27.3±2.80 ka (Ma et al., 2018). Besides, the deformed layers in the Shawan section (26.5-24.1 ka; Wang et al., 2011; Wang et al., 2012), and the convolution structure in Haizipo (27.74±0.47 ka; Wang et al., 2012), are all confirming that the outburst happened 27.11±0.18



ka ago. Also, the palaeo-landslide in Qiangyangqiao (26.54±0.53 ka, 27.28±0.41 ka; Wang et al., 2012),

and the palaeo-dammed lake in Maoxian (26.81±0.98 ka; Wang et al., 2007), suggest that the upper

reaches of the Minjiang River have been several disastrous events during 27 ka.

Subsequently, the palaeo-dam blocked the river again, and the elevation may have reached or

exceeded the position of the highest lacustrine deposits in Taiping Terrace. Around 17.15±0.18 cal. ka

BP, the palaeo-dam was broken, exposing the highest lacustrine deposits of the Taiping Terrace. Besides,

the palaeo-landslide in Manaoding deposited at 16.75±0.62 cal. ka BP (Wang et al., 2012), suggests that

an event did occur in ~ 17 ka, causing the second outburst event, and the formation of palaeo-landslide

the downstream.

As seen from our dating results, between $10.92 \pm 0.98$ and $10.13 \pm 0.63$ ka, the lake level of the

palaeo-lake descends from the highest point in the Taiping terrace to T1. Between $10.11 \pm 0.66$ and 10.08

$\pm 0.50$ ka, the lake level of the palaeo-lake descends from the position of T7 to T5 in the Tuanjie Terrace.

It is evident that the lake level of the palaeo-dammed lake fluctuated considerably at ~10 ka, and such a

rapid decline is considered to be a third dam outburst during this period. The third outburst event is a

gradual failure of the palaeo-dam. In addition, the lake fluctuation of palaeo-lake during ~10 ka also can

be found in the Taiping, Shawan, and Tuanjie profiles (Zhong, 2017). After that, the dam body was in a

stable state, until the fourth dam break event occurred around 9.35 ka. Finally, the present riverbed was

formed.

About 30 ka BP, during the last glacial period, the formation of Tuanjie Terrace might also have

been affected by tectonic activities (such as earthquakes) and climate changes (Shen, 2014; Wang, 2009;

Luo et al., 2019; Wang et al., 2012). To better understand the constraint of tectonic activities, climate

changes, and palaeo-dam on terrace formation, we discuss the effects of these three factors below.

### 5.4 The formation mechanisms of terraces

Plenty of studies stated that tectonic activities and climate changes played an important role in

mountainous terrace generation and landscape evolution (Maddy et al., 2005; Burgette et al., 2017; Chen

et al., 2020; Gao et al., 2020; Narzary et al., 2022; Ma et al., 2023). Recently studies have also taken

disaster events into account (Wang et al., 2021a; Yu et al., 2021). In the Diexi area, the huge thickness

(>200 m) of lacustrine deposits and the multiple loess-paleosol sequences, suggest that terraces are



subject to tectonic uplift, climate fluctuations, and damming effects. Here, we discuss the impact of these

three factors on the Tuanjie and Taiping terraces.

### 5.4.1 The reflection of terraces to tectonic activities

As the distance between Tuanjie and Taiping is only 12 km, we regard the Tuanjie and Taiping

Terraces as being in the same tectonic uplifting background. In Section 5.2.1, we divided the upper

Minjiang River into three parts, the Zhangla to Gonggaling area (upstream of the Diexi area), the Diexi

area, and the Maoxian-Wenchuan area (downstream of the Diexi area). These three parts have different

incision rates (as shown in Table. S1): (1) Gonggaling-Zhangla, the incision rates range from 0.4 to 85.3

mm/yr; (2) Shawan-Tuanjie, the incision rates vary from 3.6 to 198.0 mm/yr; and (3) Maoxian-

Wenchuan, the incision rate is 4.3-58 mm/yr. While the Minshan Block, the area around the Minjiang

River, has had an average uplift rate of 1.5 mm/yr since the Quaternary (Zhou et al., 2000).

During the damming period of the Diexi palaeo-dammed lake (32-10 ka), the incision rates of these

three sections have respectively ranged from 8.3-85.3 mm/yr, 13.6-198 mm/yr, and 58 mm/yr (Table.

S1). The large difference in incision rates in the Diexi area may be caused by the different heights and

positions of the samples, and the uncertainty due to different dating methods. As the incision rates in the

Diexi area are far more than the uplift rate of the Minshan Block, therefore, tectonic activity does not

seem a critical factor for the evolution of terraces in the Diexi area.

### 5.4.2 Climate fluctuations affected the formation of terraces

Diexi has undergone three transitions from cold and dry to warm and humid climates during 40.5-

30.0 ka (Zhang et al., 2009). Especially at 30-15 ka, Diexi had ten distinguishable climatic and

environmental periods (Wang et al., 2014), and seven alternate periods between cold to warm during

22~10 ka (Wang, 2009). In the Zhangla Basin, the climate is cold during 35-20 ka, then, changes from

cold to warm during 20-10 ka (Zhu, 2014). Tuanjie has the same trend as Zhangla Basin during 20-10

ka, and Diexi is subject to frequent climate fluctuations since 40 ka.

The chronological results of deposits range from 32.40±1.91 ka to 3.78±0.15 ka, we compared our

dating ages with variation in the climate curves (Fig. 7). Curves a, b, c, and d respectively represent the



Sanbao Caves, the $\delta^{18}O$ of Hulu Cave, the East Asian Monsoon, and the GISP 2 $\delta^{18}O$ record. These four curves have a great fluctuation from the end of the Last Glacial Maximum (LGM) to the early Holocene, and then, when entering the Holocene, an abrupt change. The two gravel units are older than the

lacustrine deposits of the terraces in which they are located, and these two gravel units were deposited during the last glacial maximum and non-climatic events, which indicates that the climate has little influence on this depositional unit. But rather that the input of materials from the upper reaches of the Minjiang River did not cease during the blockage of the palaeo-dam, and that there were sufficient materials to form such a thick unit of gravel. Loess unit was deposited on T2 at the end of the Younger

Dryas and before the warm period, which reflects the cool depositional environment. Most of the paleosol units were deposited in the early Holocene, consistent with the indicative warming condition. Thus, it appears that climate change had a greater effect on eolian sediments than on terrace formation.

The first two dam-break events (27.11±0.18 and 17.15±0.18 ka) are not related to climate change, the third and fourth outburst events (~10 and ~9.35 ka) happened during warm and humid climates. That

is, the formation of the terrace sped up during the early-Holocene period. Although the dam-break events became frequent during the early Holocene, it is difficult to confirm that warmer periods triggered an increase in rainfall, causing overtopping of the dam, forming terraces. Therefore, the lack of high-resolution rainfall data makes it difficult to specify the influence of climate on terrace formation.

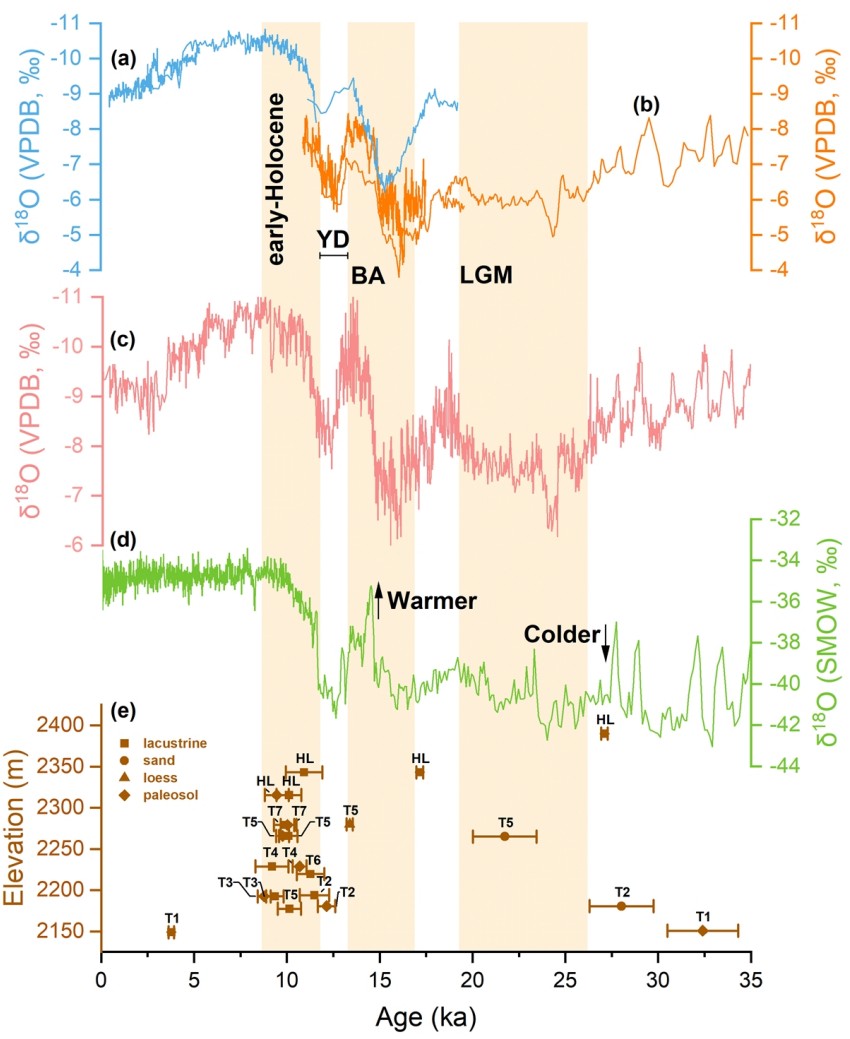

**Figure 7. Paleoclimate records compared with the Diexi OSL and AMS $^{14}$C-based terrace chronology. (a) Sanbao Caves (from Wang et al., 2008). (b) δ$^{18}$O of Hulu Cave (from Wang et al., 2001). (c) East Asian Monsoon (from Cheng et al., 2016). (d) the GISP 2 δ$^{18}$O record (from Grootes et al., 1993). (e) Ages of each terrace along the Diexi palaeo-lake. The vertical-orange bars in the figure show the duration of the early Holocene, the Bølling-Allerød interstadial, the LGM, and the YD are located between the early-Holocene and Bølling-Allerød interstadial.**

But, climate change can affect the topography of terraces. Tuanjie Terrace T2 has an irregular age-depth sequence, suggesting that the lake level dropped and rose repeatedly by 11 m during 18.60±2.86~10.63±1.27 ka (Table S1) (ages dated by Mao, 2011; Jiang et al., 2014; Shi, 2020). That is, the geomorphological features of Tuanjie Terrace T1 and T2 have been influenced by climate change. A



repeating long-term wave erosion, fluctuating along the palaeo-lake level, caused beveling and backwearing of T2 (Malatesta et al., 2022). This contributed to the Tuanjie T1 becoming the widest terrace surface (Fig. 7).

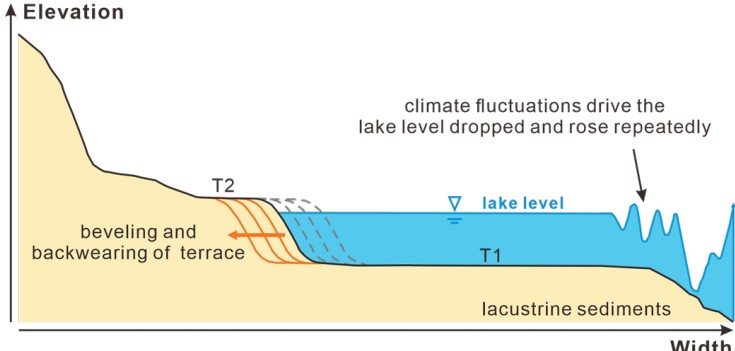


**Figure 8. Climate fluctuations drive the landscape evolution of the Tuanjie Terrace T1 and T2 (modified from Malatesta et al., 2022). Repeated drops and rises of the lake level are influenced by climate change, resulting in beveling and backwearing of the terrace, and the widest surface of T1.**

**5.4.3 The instability of the palaeo-dam**

Damming and outburst events can strongly impact both the upstream and downstream, causing aggradation and incision (Fig. 9) (Hewitt et al., 2008; Korup and Montgomery, 2008). The effects of the blocking event on the upstream and downstream are a fast rise of the water level, followed by potential upstream flooding (Guo et al., 2016). The accumulation upstream can abrade and protect the channel

bedrock and has great effects on the evolution of rivers and regional landscapes (Korup et al., 2010; Yu et al., 2021). During the blockage period, the dam impeded the incision of the river and maintained its base level. The gravity and density flowed brought materials (gravels, cobblestones, sands, etc) deposited in the Diexi palaeo-dammed lake and formed a channel. These materials eroded and deposited along the channel, formed as alluvial deposits. At the outburst period, the lake level dropped, and the river cut

through the sediments, forming terraces along the river. As each outburst event did not result in a complete breach of the palaeo-dam, the terrace was formed by the river channel was downcut after each breach (Wang et al., 2012). Then, the downstream channel restarted, a new, narrow and steep valley was formed (Wang et al., 2021a).



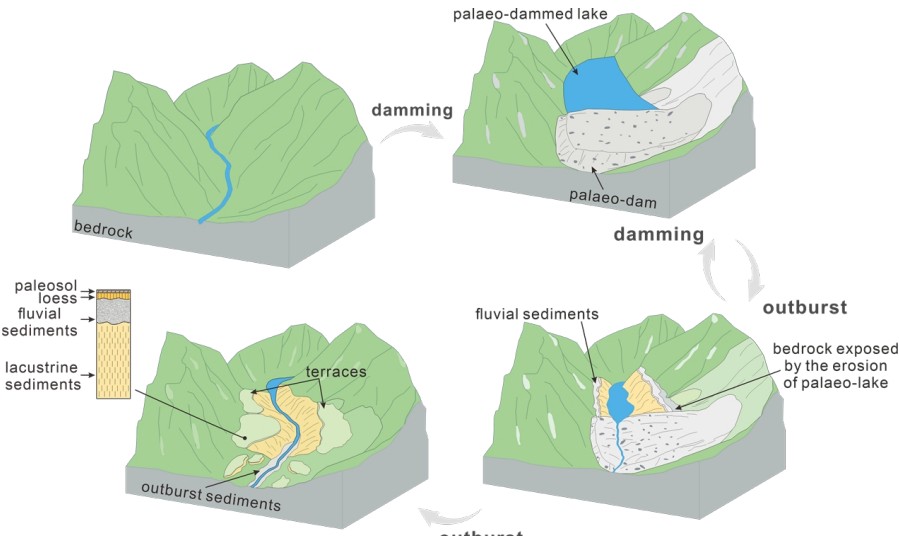


**Figure 9. Model of palaeo-landslide dam driven valley landscape and terrace development. The palaeo-landslide blocked the river and formed a palaeo-dam, the water level rose and formed a palaeo-dammed lake. During the outburst period, the palaeo-dammed lake shrank, the lacustrine and fluvial sediments were exposed, and the palaeo-dam was cut down to form a river channel. Subsequently, through repeated damming**


**and outbursts, the palaeo-dam completely collapsed and deposited as outburst sediments, and terraces were formed along the river. The stratigraphical sequence of the terrace is lacustrine sediments, fluvial sediments, loess, and paleosol, from bottom to top,.**

Most of the lacustrine sediments of the Tuanjie and Taiping Terraces were deposited during the

32.40 ±1.91 ka to 9.20±0.89 ka, and the ages at higher positions are older than the lower terraces. We

propose that this is caused by the repeated blockings and outbursts of the palaeo-dam. The largest scale

of the palaeo-dam formed at the beginning, allowing for a high lake level, and the corresponding

lacustrine position was higher and the age was older. After the palaeo-dam was broken, the height of the

palaeo-dam body dropped and the lake surface dropped, thus forming the lower terraces with younger

ages. Besides, the downcutting rates of Tuanjie T7 to Tuanjie T5 (1652.33 mm/a) and Tuanjie T4 to

Tuanjie T3 (242.33 mm/a) were higher than the maximum channel incision rate (198.00 mm/a) of study

area around 10 ka ago (Duan et al., 2002). Such rapid downcutting supported the damming and dam-

breaking of the palaeo-dam is a key factor in the formation of the Diexi terraces.

As terraces developed near the palaeo-dam are considered the remains of the dam itself (Zhang et

al., 2013; Yu et al., 2021), Tuanjie terraces reflect that Diexi palaeo-dammed lake experienced several

outbursts before its extinction, and each terrace corresponds to an outburst event (Duan et al., 2002;



Wang, 2009; Wang et al., 2020), with the interval of each outburst time is about 1500 years (Wang et al., 2007). Our dating results supported that the Diexi palaeo-landslide dam have multiple dam-breaking events, as mentioned in Section 5.3. During 32.40±1.91 ka, the height of the dam body reached at least

the highest lacustrine deposits of Tuanjie Terrace. As the age of the fluvial deposits of Tuanjie T2 (28.03±1.72 ka) is similar to that of outburst sediments (~27.3±2.8 ka, Ma et al., 2018), reveals that the first outburst event at ~27 ka, causing the lake level to drop to the surface of the Tuanjie T2, and accompanying with the input of upstream gravels deposited at the T2 position. As the ages of loess and paleosol are distributed around 13.38±0.08 to 3.78±0.15 ka, supported that these have been deposited

after the third outburst event.

       So, the evolution of each terrace and paleo-dam can be summarized as follow: a palaeo-dam was formed before 32.40±1.91 ka, and its tip reached above 2390 m (Fig. 10a). Around 27.11±0.18 ka ago, the palaeo-dam burst open, exposing the Tunajie T2 and the highest lacustrine deposits of Tuanjie (Fig. 10b). Then, the palaeo-dam blocked the river again between 27.11±0.18 and 17.15±0.18 ka ago. At this

period, the lake shore receded till the Taiping Terrace (Fig. 10c). The second palaeo-dam outburst happened 17.15±0.18 ka ago, exposing the highest lacustrine deposits of Taiping (Fig. 10d). Tunajie T7 to T5 have been formed during the third dam-breaking, around 10 ka, which was a gradual outburst event (Fig. 10e). At last, during the fourth outburst of ~9.35 ka ago, Tuanjie T4, T3 and T1 terraces formed, and eolian sediments were deposited on terraces during this period (Fig. 10f).


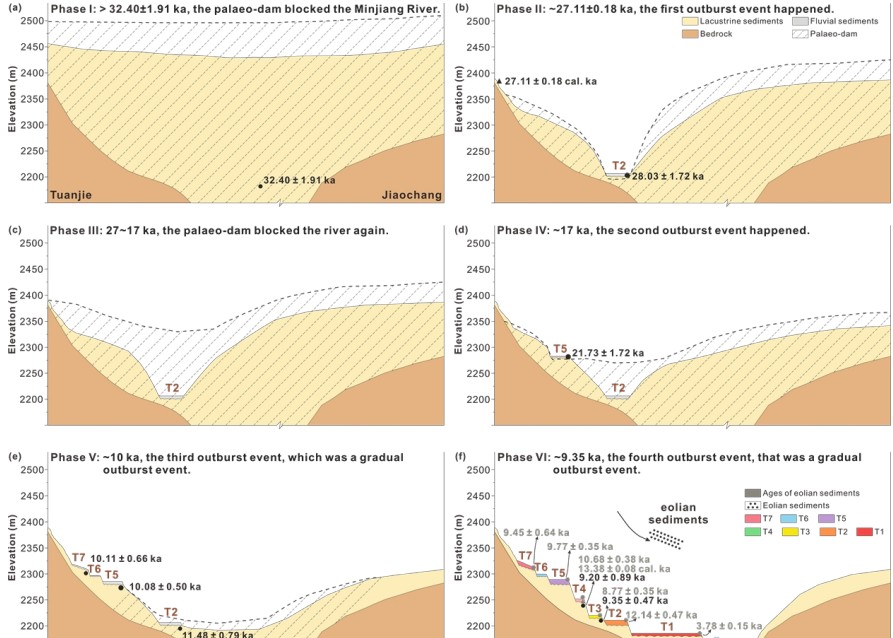

**Figure 10. Schematic evolution and the relationship between palaeo-landslide dam and Tuanjie Terrace. (a)**
**Before 32.40±1.91 ka, the palaeo-dam blocked the Minjiang River, and lacustrine sediments were deposited.**
**(b) The first outburst event happened at ~27.11±0.18 ka. The lake level dropped to the surface of Tuanjie T2,**
**and the input of upstream gravels deposited at the T2 position. During this period, the lacustrine sediments**
**of T2 and the highest positions were exposed. (c) The palaeo-dam blocked the river again at 27~17 ka, the**
**palaeo-lake shore receded till the Taiping. (d) The second outburst event happened at ~17 ka, and the highest**
**lacustrine deposits of Taiping Terraces were exposed. (e) The third outburst event occurred at ~10 ka, which**
**was a gradual outburst event. Terrace T7, T6, and T5 were formed during the third dam-breaking event.**
**Terrace T2 was influenced by the repeatedly dropped and rose of palaeo-lake level during this period, as**
**discussed in Fig. 8. (f) The fourth outburst event, a gradual outburst event happened at ~9.35 ka. Terraces**
**T4, T3, and T1 were exposed, and eolian sediments were deposited during this period. The detailed ages are**
**shown in Fig. 5.**

**6 Conclusions**

Tuanjie and Taiping Terraces have similar stratigraphic sequences with a base of lacustrine deposits,
overlain then by gravels, loess, and paleosol. Gravels of the Tuanjie terraces have been carried by the
Minjiang River, while those of the Taiping terraces are influenced by the Luobogou Gully. Two
sequences of mud-clast layers in the Taiping T3 terrace implied that the formation of the T3 terrace is
controlled by damming events.



This combined observation of geomorphology, sedimentology, and chronology shows that Taiping terraces T1 to T3 are matched with Tuanjie T5 to T7. Tectonic movement and climate fluctuations are not the main factors of terrace formation; damming and breaking events, instead, play an important role. With the measurements, two damming and four outburst events have been identified. About 32.40±1.91

ka, the river was blocked, and the lake level rose to the highest level recorded by the lacustrine deposits. The dam was intact until 27.11±0.18 ka, when the first outburst event happened: in that instance, the height of the palaeo-dam dropped to near the Tuanjie T2 surface. The palaeo-dam was blocking the river again between 27.11±0.18 to 17.15±0.18 ka, allowing the lake surface to extend till the Taiping Terrace. It then broke again around 17.15±0.18 ka, exposing the highest lacustrine deposits. Tuanjie T7 to T5 are

corresponding to the third dam-breaking period, around 10 ka, it was a gradual outburst event. The Tuanjie T4, T3, and T1 are related to the fourth gradual collapse event at ~9.35 ka.

This finding has important implications for revealing the formation and evolution of the Diexi palaeo-landslide dammed lake. This knowledge is crucial to understand the formation of these terraces and reconstructing the evolution of the Diexi palaeo-landslide dam. This study proposes a new

perspective on terrace formation in the eastern margin of the Tibetan Plateau, which can help to better understand the effect of landslide dams over fluvial evolution. Besides, it has important implications in the study of the evolution of palaeo-climate and palaeo-environment, providing an insight into future mountainous engineering projects.

**Author contributions**

JL wrote the manuscript and analyzed the data. XF and ZD discussed the results and provided guidance and funding. ML polished the language.

**Competing interests**

An author is a member of the editorial board of the journal Earth Surface Dynamics. The peer-

review process was guided by an independent editor, and the authors have also no other competing interest to declare.



**Acknowledgments**

We thank Lanxin Dai, Chengbin Zou, Yujin Zhong, Binbin Luo, Bing Xia, Kunyong Xiong for

fieldwork assistance, and Xiangyang Dou for revising the figures.

**Financial support**

This research is financially supported by the Funds for National Science Foundation for Outstanding

Young Scholars, Grant no. 42125702, the National Natural Science Foundation of China, Grant no.

42207223, and the Natural Science Foundation of Sichuan Province, Grant no. 2022NSFSC003.

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
