# Peer review of "Palaeo-landslide dams controlled the formation of Late Quaternary terraces in Diexi, the upper Minjiang River, eastern Tibetan Plateau"

_EGUsphere, 2023_

## Author Comment (AC1)

**The Response to Comments from Anonymous Reviewer**

| General Comments |
|---|
| The manuscript has some linguistic deficiencies, particularly translation issues, which are discussed in more detail in the technical corrections. To improve the flow of reading, it is recommended to summarize some of the many short sentences. It is advisable to have a native speaker proofread the manuscript.

The abstract provides a concise summary of the manuscript. The discussions are not well structured and difficult to follow, while the summary is more clear.

The manuscript may be accepted after major revision, based on the following comments. |
| **Response**

We appreciate you very much for your positive and constructive comments on our manuscript. We have fully revised our manuscript and have addressed all of your comments. All the revisions have been addressed in the revised manuscript shown in red. The manuscript has been revised by native English speaker to improve the grammar problems and readability, and clarify our ideas. |
| **Specific comments** |
| **Comment 1**

L15: the detail about the mud-phyllite in T3 is not of interest in the abstract. |
| **Response 1**

Thanks for your comment. We deleted this sentence in the abstract. |
| **Comment 2**

L21: You are rounding every age in this paragraph except for 9.35ka. |
| **Response 2**

We would like to thank you for pointing out this issue.

Considering these ages obtained from the OSL method, we rounded the ages of all phases as:

Phase I is 32 ka, Phase II is 27 ka, Phase III is 27~17 ka, Phase IV is 17 ka, Phase V is 10 ka, and Phase VI is 9 ka. |
| **Comment 3**

L39: Please mention the studies. |
| **Response 3** |

Thanks for your comment. We have cited relevant studies on *L41-43*.

Currently, there are few studies on the influence of disaster events on the formation and evolution of terraces *(Chen et al., 2016; Hu et al., 2018; Montgomery et al., 2004; Xu et al., 2020; Yuan and Zeng, 2012; Zhu et al., 2013)*, and further exploration is advisable.

**Comment 4**

L64-65: "might need to be further studied" and "should be considered" appears indecisive.

**Response 4**

Thanks for your comment.

We modified this sentence: "*Due to the lack of sedimentary sequence and chronological data, further study is needed on the evolution of palaeo-dam and the causes of terrace formation. The roles of tectonic activity, climate, river blockage and outburst events are crucial for discussing the formation of terrace staircases.*" on *L67-70.*

**Comment 5**

L95: Do you mean "alpine erosion landform" ?

**Response 5**

Thanks for your comment. Yes, we mean "alpine erosion landform", and we modified it on *L101,* as follows:

It has a typical *alpine erosion landform* with an 1868-4800 m elevation.

**Comment 6**

Fig 1: The symbols on the map for the study area and county are not clear. It is difficult to understand the geological formations from the map. The villages on the map c & d might not be at their correct locations, a directional arrow can help here.

**Response 6**

Thanks for your comment.

We bolded the lines in Figure 1b to clarify the boundaries between the strata. We changed the font color for Xiaohaizi and Dahaizi, and added arrows to indicate their locations. Additionally, we added three inferred faults, which we overlooked before. In Figures 1c and 1d, the term "village" is replaced with "Terrace" to better reflect the theme of this manuscript.

[Figure]

**Comment 7**

L122-123: This assumes that the terrace levels are increasingly younger the higher they are, which is not the case. I suggest not using the phrases "oldest" and "youngest".

**Response 7**

Thanks for your comment. We sincerely appreciate the significant suggestions. We deleted this description, and rewrote it on *L127-128*, as follows:

*These terraces are named in order of Terrace 1 (T1) to Terrace 7 (T7) from bottom to top.*

**Comment 8**

L139: Why did you take OSL samples from different units? In my opinion, this immediately introduces a problem of water content and dose rate. It is good as an age check next to another sample from the same terrace, but I find it a bit difficult this way. And what about T6?

**Response 8**

Thanks for your comment. We are sorry that this part was not clear in the original manuscript.

We collected dating samples from the top of lacustrine deposits, and gravel units, and the bottom of loess and paleosol units. The dating of lacustrine deposits confirms the damming process of the palaeo-landslide dam, the dating of gravel units corresponds to the outburst time, and the dating of loess and paleosol units used to determine the time of terrace geomorphic stability. Therefore, we believe it is necessary to conduct dating for each unit. We added a description on *L137-140, in Sect 3.2*, as follows:

*To clarify the damming and outburst processes of the palaeo-dam, and the stability time of terraces, we collected samples from the top of lacustrine and gravel units, and the bottom of loess and paleosol units.*

The ages of each phase (Phase I-VI) are determined based on the ages of lacustrine deposits. We did not compare the ages of different sedimentary units. Thus water content and dose rate do not affect our results.

The terrace T6 has experienced significant deformation, making it difficult to obtain suitable samples. Therefore, we did not collect samples from this terrace.

**Comment 9**

L156: Did you perform a density separation prior to etching to separate the quartz from the other material?

**Response 9**

Thanks for your comment. We are sorry that this part was not clear. We rewrote this process on *L154-166 in Section 3.2.1 OSL dating*, which described how to separate the quartz from the other materials, as follows:

*Samples were processed and measured at the Institute of Earth Environment, Chinese Academy of Sciences. The quartz grains were extracted following the laboratory pre-treatment procedures (Kang et al., 2020; Kang et al., 2013).The sediments at the two ends of the tubes, which may be exposed to daylight during sampling, were removed. And, the unexposed samples were prepared for equivalent dose ($D_e$) and environment dose rate determination. Approximately 50 g samples were treated with 30% HCl and 30% $H_2O_2$ to remove carbonates and organic matter, respectively. Then, the samples were washed with distilled water until the pH value of the solution reached 7. For samples IEE5542 and IEE5550, the coarse fractions (90-150 μm) were sieved out and etched with 40% HF for 45 mins,*

*followed by washing using 10% HCl and distilled water. For the other 17 samples, the fine polymineral grains (4-11 μm) were separated according to the Stokes' law. These fine polymineral grains were immersed in 30% $H_2SiF_6$ for 3-5 days in an ultrasonic bath to extract quartz.Finally, the purified fine (coarse) quartz was deposited (mouted) on stainless steel discs with a diameter of 9.7 mm for experimental use. The purity of quartz was verified by IRSL intensity and OSL IR depletion ratio (Figs. S1 and S2a; Duller, 2003).*

Because of adequate purity of quartz after HF etching of coarse polyminerals (Figs. S1 and S2), we did not perform a density separation after HF etching.

**Comment 10**

L162: What is the exact protocol you have been using?

**Response 10**

Thanks for your comment. The protocol illustrated on *L171-172*, as follows:

*The single-aliquot regenerative-dose (SAR) protocol (Table S1; Murray and Wintle, 2000; Wintle and Murray, 2006) was utilized to determine the Equivalent Dose ($D_e$), as used in Kang et al. (2020).*

**Comment 11**

L166: How did you measure the environmental dose rate? Did you take appropriate samples in the field?

**Response 11**

Thanks for your comment. Sorry that this part was not clear. The samples of environmental dose rate are separated from the OSL samples, and they were not specifically collected in the field. And the environmental dose rate determination was presented on *L156-158* and *L181-L191*, as follows:

*…The sediments at the two ends of the tubes, which may be exposed to daylight during sampling, were removed. And, the unexposed samples were prepared for equivalent dose (De) and environment dose rate determination….*

*The environmental dose rate was estimated from the radioisotope concentrations (uranium, thorium, and potassium) and cosmic dose rates. U and Th concentrations were determined by inductively coupled plasma mass spectrometry (ICP-MS), while K concentration was measured by inductively coupled plasma optical emission spectrometry (ICP-OES). The cosmic dose rates were calculated using the equation proposed by Prescott and Hutton (1994). The α-value of fine (4-11 μm) grained*

*quartz was assumed to be 0.04±0.002 (Rees-Jones, 1995). Considering the current climate conditions, the sedimentary facies, and past climate changes since the sample deposition, the water content of the gravel and paleosol was assumed to be 10±5%, while the water content of lacustrine deposits was estimated to be 20±5%. Dose rate was calculated using the Dose Rate and Age Calculator (DRAC) (Durcan et al., 2015). Finally, the quartz OSL ages were obtained by dividing the measured $D_e$ (Gy) by the environmental dose rate (Gy/ka).*

**Comment 12**

L170: Why did you sample there? In general, your choice of sampling location (also for OSL) is not entirely clear to me.

**Response 12**

Thanks for your comments. We added the description to explain why we collected these samples on *L196-201* as follows:

*The AMS [14]C sample collected from the top of the Taiping Terrace was used for comparison with the OSL sample (TP19-1), which was taken from the same position. The AMS [14]C sample collected from the top of the Tuanjie Terrace was compared with the AMS [14]C dating of the top of the Taiping Terrace. Utilizing the same dating method for age comparison enhances credibility. Field investigations showed that the loess unit of the Tuanjie T4 was the most complete and easier to collect, therefore, we collected the loess sample from T4.*

**Comment 13**

Fig 4.: This is a very nice and vivid illustration.

**Response 13**

Thanks for your comment.

**Comment 14**

L236: Is there an explanation for why T5 and T7 are missing the loess unit?

**Response 14**

Thanks for your comment. We added the description on *L269, in Sect 4.2.1*, as follows:

*The absence of loess units in T5 and T7 may be caused by erosion and human activities.*

**Comment 15**

Fig 5.: The x-axis font is too small

**Response 15**

Thanks for your comment. We magnified the x-axis font of Figure 5 and corrected a mistake made during the drawing process. We modified the lithofacies code of the gravel layer of T3 from *Gh* to *Gci*. As it is an inverse grading, this correction was necessary.

The description of the lithostratigraphy of Tuanjie T3 has also been modified on *L258-261*, as follows:

*In Tuanjie T3 (Gci), the gravel units are poorly sorted and sub-circular to round gravels with a 3-25 cm diameter and exhibit inverse grading. These features suggest that the gravel units of T2 and T3 are clast-rich debris flows with high strength or pseudoplastic debris flows with low strength.*

[Figure]

**Comment 16**

L272: It is not "Optional", but "Optically Stimulated Luminescence"

**Response 16**

We are really sorry for our careless mistakes. Thank you for your reminder. Considering that the full term has been mentioned as "optically stimulated luminescence (OSL)" on *L73*, we used the

abbreviation "OSL" directly in this sentence. We modified it on *L308*, as follows:

*OSL* dating of lacustrine deposits in Tuanjie terraces yielded ages of 32.40±2.07 ka for the T1, 10.92±1.01 ka for the T2, 9.46±0.54 ka for the T3, 7.97±0.81 ka for the T4, 10.36±0.61 ka for the T5 and 9.98±0.77 ka for the T7.

**Comment 17**

L281/282: Your terraces do not become younger with increasing elevation. T5 & T7 are older, but higher than T3 & T4 for example. Generally, this section is very difficult to follow (L272-286).

**Response 17**

Thanks for your comment. We deleted this sentence, and summarised on *L312-314*, as follows:

*The chronological results of lacustrine deposits are chaotic. Tuanjie T1-T4 becomes younger with increasing elevation. Tuanjie T5 and T7 have a similar age, but are older than T3 and T4. The highest lacustrine deposits are only about 5 ka younger than T1.*

**Comment 18**

L307: Have you done a bleaching test to correct the residuals?

**Response 18**

We rewrote the bleaching extent on *L335-337*, as follows:

*Considering the fine silt dominated nature, the relatively stable depositional environment, and the normal distribution of $D_e$ particularly for the two coarse samples, we assume that all the OSL samples were well bleached before deposition.*

**Comment 19**

L336: Round the numbers as they suggest a level of accuracy that you don't have. It is unclear what you are referring to with Table S1. You must mention the source for these ages.

**Response 19**

Thanks for your professional suggestions. We rounded these numbers as "830 ka", "1 ka", "40 ka", "6 ka" on *L372*, as follows:

In summary, the terrace ages along the upper Minjiang River span from *830* to *1* ka, with the majority formed between *40* and *6* ka. The Diexi area shows a higher concentration of terraces than the upstream and downstream regions, with these terraces primarily formed from *30* to *0* ka.

We mention the source of these ages on *L364-369*, as follows:

The ages of the upstream terraces indicate that the formation and evolution of terraces in the upper Minjiang River began around 830 ka (the early Pleistocene, *Zhao et al., 1994*), and primarily formed between 47-2 ka (Fig. 6). The terraces in the Diexi area have ages that are distributed between 550 and 50 ka *(Duan et al., 2002; Guo, 2018; Kirby et al., 2000; Wang et al., 2020; Wang et al., 2007; Wang, 2009; Yang et al., 2003; Zhong, 2017; Gao and Li, 2006; Jiang et al., 2014; Luo et al., 2019; Mao, 2011; Zhang, 2019)*, with the majority formed between 32-2 ka (Fig. 6). Downstream terraces were deposited between 400 and 50 ka *(Yang et al., 2003; Yang, 2005; Zhao et al., 1994; Zhu, 2014)*, with a significant portion formed between 40 to 20 ka (Fig. 6).

**Comment 20**

L427-433: You have large ranges for incision rates here and you claim that there are significant differences. However, the differences are mainly in one area only, rather than compared to all of them.

**Response 20**

Thanks for your comment. Sorry for the misunderstanding. This part is not clear, "Diexi area" on *L433* (previous version) means "Taiping-Tuanjie".

We rewrote this part on *L458-470*, as follows:

*Considering the short distance of only 12 km between Tuanjie and Taiping, we regard them as in the same tectonic uplifting background. In Section 5.2, we divided the upper Minjiang River into three parts: the Zhangla to Gonggaling area (upstream of the Diexi area), the Diexi area (Taiping-Tuanjie), and the Maoxian-Wenchuan area (downstream of the Diexi area). During the damming period of the Diexi palaeo-dammed lake (32-10 ka), the incision rates in these three sections ranged from 8.3-85.3 mm/yr, 13.6-198 mm/yr, and 58 mm/yr, respectively, from upstream to downstream (Table. S2). And the Minshan Block, which includes the Minjiang River, has experienced an average uplift rate of 1.5 mm/yr since the Quaternary (Zhou et al., 2000). It can be observed that the incision rates of the upper reaches of the Minjiang River during the period of 32-10 ka are significantly higher than the uplift rate of the Minshan Block, indicating that tectonic activity has little influence on the formation of regional terraces. In particular, the Taiping-Tuanjie region has a higher incision rate than the upstream and downstream areas, highlighting its unique characteristics. That is, tectonic activity is not a critical factor in the evolution of Tuanjie and Taiping terraces.*

**Comment 21**

L510: The results now indicate the exact opposite of the previous findings (see L122 & L281)

**Response 21**

Thanks for your comment. We deleted and rewrote this sentence on *L541-542*, as follows:

*Most lacustrine deposits in the Tuanjie and Taiping Terraces were deposited from 32.40 ±2.07 ka to 7.97±0.81 ka.*

**Comment 22**

Fig. 10: Really nice presentation of the process and absolutely necessary.

**Response 22**

Thanks for your comment. We round the phase ages in this figure:

[Figure]

**Technical corrections**

**Comment 1**

L27: Word repetition "evolution".

**Response 1**

Thanks for your comment. We modified this sentence as "*Terraces, as a natural archive of the process of valley evolution, are used to explore the controlling mechanisms of river landscapes*" on *L29-30*.

**Comment 2**

| |
|---|
| L46: "The upper Minjiang River is located in the eastern Tibetan Plateau, and it is characterised by a wide distribution of three-tiered terraced." |
| **Response 2**
 We modified it on *L49-50*, as follows:
 *The upper Minjiang River is located in the eastern Tibetan Plateau, and a wide distribution of three-tiered terraces characterizes it.* |
| **Comment 3**
 L56-58: Word repetition "sedimentary system". It can be summarized as "fluvial, lacustrine, alluvial fan [...] sedimentary system". |
| **Response 3**
 Thanks for your comment. We rewrote this sentence: "*The analysis of lithofacies and sedimentary systems determined that the Diexi area is mainly composed of fluvial, lacustrine, alluvial fan and eolian sedimentary systems*" on *L58-59*. |
| **Comment 4**
 L61: "This indicates that" instead of "This is". |
| **Response 4**
 We modified it on *L63*, as follows:
 *This indicates that* the Diexi palaeo-dammed lake has experienced at least one outburst flood event |
| **Comment 5**
 L74: Colloquial. "The Diexi area is located in the upper reaches of the Minjiang River". And please connect the first two sentences. |
| **Response 5**
 We rewrote the first sentence on *L79-81,* as follows:
 *The Diexi area is located in the upper reaches of the Minjiang River, which belongs to the northeast margin of the Tethys Himalayan domain and the Barkam formation zone, on the eastern margin of the Bayan Har Block (Fig. 1a).* |
| **Comment 6**
 L78: "..and the steep slopes on both sides of the river valley have a gradient of 30-35°." |
| **Response 6** |

We modified it on *L83*, as follows:

…and *the steep slopes* on both sides of the river valley have a gradient of 30-35°.

**Comment 7**

L85-86: Word repetition "about".

**Response 7**

We modified on *L90-92,* as follows:

*The highest elevation of the palaeo-landslide is 3390 m, and the main slide direction is SW18°. The length and width of the palaeo-landslide are respectively about 3500 m and 3000 m, with a volume of the accumulation reaching 1.4 to 2.0×10⁹ m³ (Zhong et al., 2021).*

**Comment 8**

L97-98: "The climate of the entire region is monsoonal, being influenced by the Plateau Monsoon, the Westerlies, and the East Asian Monsoon."

**Response 8**

We modified it on *L103-104*, as follows:

*The climate of the entire region is monsoonal, being influenced by the Plateau Monsoon, the Westerlies, and the East Asian Monsoon.*

**Comment 9**

L194: The heading repeats. You should either write introductory words to the following chapters under chapter 4.1. or simply omit the top chapter. Same for chapters 5.2 and 5.2.1

**Response 9**

Thanks for your comment. We omitted the "4.1 Terraces distribution and sequence", and modified *Section 4.1.1* to *Section 4.1*, *Section 4.1.2* to *Section 4.2*. We rewrote "Tuanjie Terrace" as "4.2.1 Tuanjie Terrace", and "Taiping Terrace" as "4.2.2 Taiping Terrace".

We deleted "5.2 Evolution of terraces in the upper Minjiang River", and modified *Section 5.2.1* to *Section 5.2* and *Section 5.2* to *Section 5.3*.

We also renamed *Section 5.5* as "*The formation and evolution mechanisms of terraces*".

**Comment 10**

L199: Word repetition "extension/extends"

**Response 10**

We rewrite this sentence as "*Terrace T1 has the most significant extension towards the center of the Diexi Lake.*" on *L227-228*.

**Comment 11**

L200: "On a high mountain" is colloquial.

**Response 11**

Thanks for your comment. We used "on the hillside" instead of "on a high mountain" on *L228*, as follows:

Taiping terraces developed *on the hillside* with a slope of 40°-60°, influenced by landslides and croplands. The horizontal extensions of T1, T2, and T3 are equal to 520 m, 380 m, and 190 m, respectively.

**Comment 12**

L231: "Angular phyllites occur in T3."

**Response 12**

We modified it on *L263*, as follows:

*Angular phyllites occur in T3.*

**Comment 13**

L332: Just write 830 ka

**Response 13**

We modified it on *L365*, as follows:

The ages of the upstream terraces indicate that the formation and evolution of terraces in the upper Minjiang River began around *830 ka (the early Pleistocene, Zhao et al., 1994)*, and primarily formed between 47-2 ka (Fig. 6).

**Comment 14**

L334: "The terraces in the Diexi area have ages that are distributed between 550-50 ka (Table S1), with the majority observed between 32-2 ka."

**Response 14**

Thanks for your comment. We modified it on *L366-369*, as follows:

The terraces in the Diexi area have ages that are distributed between *550 and 50 ka* (Duan et al., 2002; Guo, 2018; Kirby et al., 2000; Wang et al., 2020; Wang et al., 2007; Wang, 2009; Yang et al., 2003;

Zhong, 2017; Gao and Li, 2006; Jiang et al., 2014; Luo et al., 2019; Mao, 2011; Zhang, 2019), with the majority observed between *32-2 ka* (Fig. 6).

**Comment 15**

L345: "It can be seen that the terrace formation mechanism downstream is different from that upstream."

**Response 15**

Thanks for your comment. We modified it to "*These results indicate that the terrace formation mechanism downstream differs from that upstream.*" on *L381-382*.

**Comment 16**

L346: I would suggest not to write "publish".

**Response 16**

Thanks for your comment. We rewrote this sentence: "*However, sufficient evidence has not been presented to support this perspective. In the following sections, we will present additional evidence to explore this phenomenon further.*" on *L382-383*.

**Comment 17**

Fig 6: You have the same sentence here with the age 46.40 ka to 2.81 ka.

**Response 17**

Thanks for your comment. For clearly, we modified the figure name as "*Frequency distribution histogram of terrace ages since 50 ka in the upper reaches of the Minjiang River*" on *L380-381*.

**Comment 18**

L370-373: Rewirte the sentences. Not "Tx are...".

**Response 18**

Thanks for your comment. We rewrote these sentences on *L404-406*, as follows:

*Ages of the lacustrine deposits of Taiping T1 (9.46±0.99 ka) and Tuanjie T5 (10.36±0.61 ka), as well as Taiping T3 (9.93±0.75 ka) and Tuanjie T7 (9.98±0.77 ka) (Table. 3), are similar, which confirms from a chronological perspective that the two terraces correspond to each other (Fig. 5).*

**Comment 19**

Fig. 7: Use different colors with each symbol.

**Response 19**

We are very grateful for your careful reading. We modified the symbol colors, and added the "circle" and "triangle" symbols to represent the OSL dating and AMS $^{14}$C dating methods, respectively.

[Figure]

**Comment 20**

L487-489: "The upstream and downstream effects of the blockage are a rapid rise in water level followed by potential upstream flooding."

**Response 20**

Thanks for your comment. We modified it on *L521-522*, as follows:

*The upstream and downstream effects of the blockage are a rapid rise in water level followed by potential upstream flooding.*

**Comment 21**

| L492: "Gravity and density caused the material to be deposited in the palaeo-dammed Diexi Lake and formed a channel." |
|---|
| **Response 21** |

[revised manuscript text omitted]

---

## Author Comment (AC2)

| **General Comments** |
|---|
| Based on geomorphology, sedimentology, and chronology, the manuscript reconstructing two damming and four outburst events occurred in the minjiang river during the late Pleistocene, which suggests that the blockage and collapse of the palaeo-dam have been a major factor in the formation of tectonically active mountainous river terraces, and tectonic movement and climatic fluctuations, on the other end, play a minor role. I think the topic of the paper is interesting. However, this manuscript mainly focuses on sediment dating and geomorphological interpretation. The content association with dynamics is much weak, thus I am wondering whether it is suitable for this journal or not. Considering that this manuscript still has a plenty of problems, we suggest a major revision. |

**Response**

We are very grateful for your professional suggestions on our manuscript. We have considered the comments carefully and tried our best to revise the manuscript accordingly. Our responses are given in a point-by-point manner below. All the revisions have been addressed in the revised manuscript shown in red. We have asked native English speakers to improve the language, and clarify our ideas.

| **Specific Comments** |
|---|
| **Comment 1** |
| Language should be improved by a native English speaker, please avoid all kinds of grammar errors and please use scientific language to write the paper. |

**Response 1**

We are sorry for the grammatical problems and have corrected them based on your suggestions. We have asked native English speakers to polish and modify the manuscript.

| **Comment 2** |
|---|
| The evolution of Diexi landslide-dammed lake in eastern Tibetan Plateau has gotten special attention of a plenty of scientist. The author should introduce the details of Diexi landslide dammed lake, including the history of repeated landslide damming. |

**Response 2**

Thanks for your comments. We added the description on *L64-67*, as follows:

*Moreover, the sedimentological analysis also suggests that the Diexi palaeo-dammed lake*

*experienced at least two periods of blocking and outburst events (Yang, 2005; Yang et al., 2008), and four periods of fluvial progradation (Xu et al., 2020).*

The concept of repeated blockage is introduced in this manuscript, as previous studies have only mentioned multiple instances of progressive breaching. As mentioned in *L61-64*:

Currently, Tuanjie Terrace is thought to have resulted from the outburst of a palaeo-dammed lake 15000 years ago, and each terrace corresponds to different stages of outburst (Duan et al., 2002; Wang et al., 2005; Wang, 2009; Zhu, 2014).

**Comment 3**

Ordinarily, uncertainties in estimating water content during burial are one of the largest sources of uncertainty in luminescence dating methods. Please give out the process of the determination of water content.

**Response 3**

Thanks for your comment. We were in an emergency situation in the field at that time and did not have enough time to excavate water content samples that could accurately reflect the true conditions. However, considering factors such as modern climate, depositional facies, and climate change history since deposition, we assumed the water content data.

We rewrote the water content on *L186-189*, as follows:

*Considering the current climate conditions, the sedimentary facies, and past climate changes since the sample deposition, the water content of the gravel and paleosol was assumed to be 10±5%, while the water content of lacustrine deposits was estimated to be 20±5%.*

**Comment 4**

For OSL dating, the results of recycling ratios and recuperations should be presented, and aliquots with recycling ratios out of 0.9-1.1 and recuperations higher than 5% should be rejected.

**Response 4**

Thanks for your constructive comments. We added the result on *L177-180*, and presented the recycling ratios and recuperation ratios on the supplementary, as follows:

*Conventional tests in SAR protocol, including recuperation ratio, recycling ratio, quartz OSL brightness and fast-component dominated nature, growth curve shape, and $D_e$ distribution (Figs. S2 and S3), indicate that the protocol can be robustly used to date the samples in this study.*

[Figure]

*Fig. S2. Quartz OSL IR depletion ratio (with IR/without IR **a**), recuperation ratio (recuperated/natural, **b**), and recycling ratio (repeated/regenerated, **c**) for all the 222 aliquots (used for $D_e$ determination) of the 19 luminescence samples.*

[Figure]

*Fig. S3. Quartz OSL $D_e$ determination for sample IEE5543. (**a**) Natural and regenerative-dose OSL decay curves from one of the 15 aliquots used for $D_e$ determination. (**b**) Dose-response curve and $D_e$ determination for the aliquot in (**a**). (c) Probability density distribution of $D_e$ and mean $D_e$.*

**Comment 5**

The authors should select appropriate preheat and cutheat temperatures based on a preheat plateau test (Murray and Wintle, 2000) or following some case studies, and add some references.

**Response 5**

Thanks for your comment. We rewrote this progress on *L173-174,* as follows:

*Quartz grains were preheated at 260°C for 10 s for natural and regenerative-dose, and a cut-heat at 220°C for 10 s was applied for test dose.*

We rewrote and added some references in *Section 3.2.1* to clarify the OSL procedures.

*Samples were processed and measured at the Institute of Earth Environment, Chinese Academy of Sciences. The quartz grains were extracted following the laboratory pre-treatment procedures (Kang et al., 2020; Kang et al., 2013). The sediments at the two ends of the tubes, which may be exposed to daylight during sampling, were removed. And, the unexposed samples were prepared for equivalent dose ($D_e$) and environment dose rate determination. Approximately 50 g samples were treated with 30% HCl and 30% $H_2O_2$ to remove carbonates and organic matter, respectively. Then, the samples were washed with distilled water until the pH value of the solution reached 7. For samples IEE5542 and IEE5550, the coarse fractions (90-150 μm) were sieved out and etched with 40% HF for 45 mins, followed by washing using 10% HCl and distilled water. For the other 17 samples, the fine polymineral grains(4-11 μm)were separated according to the Stokes' law. These fine polymineral grains were immersed in 30% $H_2SiF_6$ for 3-5 days in an ultrasonic bath to extract quartz. Finally, the purified fine (coarse) quartz was deposited (mouted) on stainless steel discs with a diameter of 9.7 mm for experimental use. The purity of quartz was verified by IRSL intensity and OSL IR depletion ratio (Figs. S1 and S2a; Duller, 2003).*

*All OSL measurements were performed on a Lesxyg Research measurement system, with blue light at (458±10) nm, and infrared light at (850±3) nm for stimulation and a $^{90}S/^{90}Y$ beta source (~0.05 Gy/s) for irradiation. Luminescence signals were detected by an ET 9235QB photomultiplier tube (PMT) through a combination of U340 and HC340/26 glass filters.*

*The single-aliquot regenerative-dose (SAR) protocol (Table S2; Murray and Wintle, 2000; Wintle and Murray, 2006) was utilized to determine the Equivalent Dose ($D_e$), as used in Kang et al. (2020). Quartz grains were preheated at 260°C for 10 s for natural and regenerative-dose, and a cut-heat at*

*220°C for 10 s was applied for test dose. The quartz was stimulated for 60 s at 125°C with blue LEDs. The OSL signal was calculated as the integrated value of the first 0.5 s of the decay curve minus the integrated value of the last 0.5 s as the background. For $D_e$ determination, approximately 10 aliquots were measured for each sample. And, the mean $D_e$ value of all aliquots was used as the final $D_e$ value. Conventional tests in SAR protocol, including recuperation ratio, recycling ratio, quartz OSL brightness and fast-component dominated nature, growth curve shape, and $D_e$ distribution (Figs. S2 and S3), indicate that the protocol can be robustly used to date the samples in this study.*

*The environmental dose rate was estimated from the radioisotope concentrations (uranium, thorium, and potassium) and cosmic dose rates. U and Th concentrations were determined by inductively coupled plasma mass spectrometry (ICP-MS), while K concentration was measured by inductively coupled plasma optical emission spectrometry (ICP-OES). The cosmic dose rates were calculated using the equation proposed by Prescott and Hutton (1994). The α-value of fine (4-11 μm) grained quartz was assumed to be 0.04±0.002 (Rees-Jones, 1995). Considering the current climate conditions, the sedimentary facies, and past climate changes since the sample deposition, the water content of the gravel and paleosol was assumed to be 10±5%, while the water content of lacustrine deposits was estimated to be 20±5%. Dose rate was calculated using the Dose Rate and Age Calculator (DRAC) (Durcan et al., 2015). Finally, the quartz OSL ages were obtained by dividing the measured $D_e$ (Gy) by the environmental dose rate (Gy/ka).*

**Comment 6**

How did the authors obtain the final De? Please illustrate.

**Response 6**

Thanks for your comment. We illustrated on *L171-172* and *L176-180*, as follows:

*The single-aliquot regenerative-dose (SAR) protocol (Table S1; Murray and Wintle, 2000; Wintle and Murray, 2006) was utilized to determine the Equivalent Dose ($D_e$), as used in Kang et al. (2020). … For $D_e$ determination, approximately 10 aliquots were measured for each sample. And, the mean $D_e$ value of all aliquots was used as the final $D_e$ value. Conventional tests in SAR protocol, including recuperation ratio, recycling ratio, quartz OSL brightness and fast-component dominated nature, growth curve shape, and $D_e$ distribution (Figs. S2 and S3), indicate that the protocol can be robustly used to date the samples in this study.*

**Comment 7**

Line 281: "T1 has the youngest paleosol unit with an age of 3.78±15 ka. Terraces grow younger amid the increase in elevation." Obviously, this is paradoxical. Please find a better way to present the chronological results, to avoid the reader's misunderstanding.

**Response 7**

Thanks for your comments. We deleted and rewrote this sentence on *L312-314*, as follows:

*The chronological results of lacustrine deposits are chaotic. Tuanjie T1-T4 becomes younger with increasing elevation. Tuanjie T5 and T7 have a similar age, but are older than T3 and T4. The highest lacustrine deposits are only about 5 ka younger than T1.*

**Comment 8**

For the lacustrine terraces, the dates can only implied that the filling up ages of the dammed lake rather formation time of terrace. Because of this misunderstanding, the comparison between climate date and terrace "formation" ages is also problematic.

**Response 8**

Thanks for your comments.

We apologize for this misunderstanding caused by the ambiguities expressed in this text.

The Diexi palaeo-landslide dammed lake in the past 30,000 years was not a stable and continuous water storage process, as evident from our chronological data. The age of the lacustrine deposits is chaotic (Figure 5). Therefore, these ages not only represent the time when the palaeo-dam blocked the river, but also indicate the time of its failure. Specifically, at 27.11±0.18 ka, the water level of the palaeo-dammed lake reached its highest point (2900 m), but at 17.15±0.18 ka, the water surface of the palaeo-dammed lake only reached 2342.95 m (Table 4). Therefore, the dam must have failed around 27 ka in order to have another "water storage period" at a lower elevation. In other words, the palaeo-dam blocked the river between 27 and 17 ka.

During the repeated blocking and failure processes of the palaeo-dam, especially the catastrophic failure event (~27 ka, Ma et al., 2022), the banks of the river were eroded, eventually forming terrace staircases.

The comparison between climatic chronology and terrace ages in the manuscript illustrates that climatic changes have little impact on the lacustrine units, that is, they have little influence on the

process of damming and failure of the palaeo-dam. Additionally, other depositional units of the terrace did not occur under significant climatic change conditions, thus excluding the significant influence of climatic changes on the process of damming and outburst of the palaeo-dam.

**Comment 9**

"The third and fourth gradual collapse events respectively occurred at ~10 ka and ~9.35 ka." How to divide the period of dammed lake? Is it just according to the age? How can we be sure that such a small difference in results is not due to a dating error? For the optical dating of the paper, both of the method and the internal checks of the results were not sufficiently presented. The reliability of the OSL data awaits further examination and more work is needed. My biggest concern about the OSL data comes from the bleaching extent of the OSL signals before deposition. However, the problem of bleaching extent has not been mentioned and explained in this paper, not yet anyway.

**Response 9**

Thanks for your comments.

The Tuanjie terrace, as the closest terrace to the palaeo-dam, is used as a criterion for delineating the palaeo-lake period based on the chronology of lacustrine deposits and gravel units. Due to the better exposure of gravel layers in only T2 and T5 terraces, we obtained the ages of these two gravel layers. Therefore, the division of periods for the damming and failure events of the palaeo-dam is primarily based on the ages of the lacustrine deposits, combined with previous dating results. As mentioned in the text, such as *L407-425*, we listed regional paleo-disaster events during Phase I to Phase IV to support our conclusions.

In the revised manuscript, we rounded the ages to ensure data consistency:

*Phase I is 32 ka, Phase II is 27 ka, Phase III is 27~17 ka, Phase IV is 17 ka, Phase V is 10 ka, and Phase VI is 9 ka.*

For Phase V (10 ka) and Phase VI (9 ka), we initially divided them based on the ages of the lacustrine deposits. We believe that dating errors do not cause them. Firstly, our operating procedures were also standardizeds. Furthermore, climate changes during the Holocene were frequent, and the current records from archives, such as lakes and stalagmites have achieved high precision at scales of 100 years (Yuan et al., 2004; Cheng et al., 2016; etc.). Therefore, we believe that these two dam failure events should be distinguished.

We mentioned the bleaching extent on *L335-337*, as follows:

*Considering the fine silt dominated nature, the relatively stable depositional environment, and the normal distribution of $D_e$ particularly for the two coarse samples, we assume that all the OSL samples were well bleached before deposition.*

**Comment 10**

Tectonic uplift and climate changes are the two critical factors controlling the evolution of river landscapes and the formation of terraces. However, rock uplift rate is calculated from bedrock terraces, rather than lacustrine terraces.

**Response 10**

Thanks for your valuable comments.

The primary purpose of calculating the incision rate based on lacustrine deposits is to distinguish the influence of tectonic activity from the effects of damming and failure events. The main applications of uplift rate and incision rate in this study are demonstrated as follows:

The Tuanjie and Taiping terraces are located in a tectonically active area, where several thrust and inferred faults are developed within the region (Fig. 1b). To determine whether the formation of terrace staircases is related to the strong tectonic activities in the Quaternary, we compared the incision rate of the Diexi area with that of the upstream and downstream, and the average uplift rate of the Minshan Block. The results indicate that the incision rate in the upstream reaches of the Minjiang River is significantly higher than that of the Minshan Block, suggesting that the formation of terrace staircases in the region is less influenced by tectonic activity. Particularly in the Taiping-Tuanjie area, the incision rate is much higher than in the upstream and downstream areas, indicating its unique characteristics. As mentioned in *L465-470*:

*It can be observed that the incision rates of the upper reaches of the Minjiang River during the period of 32-10 ka are significantly higher than the uplift rate of the Minshan Block, indicating that tectonic activity has little influence on the formation of regional terraces. In particular, the Taiping-Tuanjie region has a higher incision rate than the upstream and downstream areas, highlighting its unique characteristics. That is, tectonic activity is not a critical factor in the evolution of Tuanjie and Taiping terraces.*

In addition, we compared the downcutting rate of the Tuanjie terraces (T7 to T5) with the river

incision rate at 10 ka in the study area. The downcutting rate of the terrace staircases was found to be significantly higher than the river incision rate. This rapid incision not only suggests that regional tectonic activity has little influence on the formation of terrace staircases, but also supports the viewpoint that the damming and breaching of palaeo-dam played a crucial role in forming the Diexi terraces. We described this in *L543-549*:

Most lacustrine deposits in the Tuanjie and Taiping Terraces were deposited from 32.40 ±2.07 ka to 7.97±0.81 ka. *Initially, the palaeo-dam had a large blocking scale, resulting in a high lake level, and corresponding lacustrine deposits at higher positions with older age. Since the palaeo-dam gradually broke, the height of the palaeo-dam body decreased, leading to a drop in the lake surface and the formation of lower staircases with younger ages, such as the age of the highest lacustrine deposits older than Tuanjie T2-T7. Besides, the downcutting rate of Tuanjie T7 to Tuanjie T5 (1652.33 mm/a) is higher than the maximum channel incision rate (198.00 mm/a) of the study area around 10 ka ago (Duan et al., 2002).* This rapid downcutting supports that the damming and dam-breaking of the palaeo-dam are critical factors in the formation and evolution of the Diexi terraces.

**References**

Cheng, H., Edwards, R. L., Sinha, A., Spotl, C., Yi, L., Chen, S., Kelly, M., Kathayat, G., Wang, X., Li, X., Kong, X., Wang, Y., Ning, Y., and Zhang, H.: The Asian monsoon over the past 640,000 years and ice age terminations, Nature, 534, 640-646, https://10.1038/nature18591, 2016.

Duan, L., Wang, L., Yang, L., and Dong, X.: The ancient climatic evolution characteristic reflected by carbon and oxygen isotopes of carbonate in the ancient barrier lacustrine deposits, Diexi, Minjiang River, The Chinese Journal of Geological Hazard and Control, 13, 91-96, https://doi.org/10.3969/j.issn.1003-8035.2002.02.019, 2002.

Duller, G. A. T.: Distinguishing quartz and feldspar in single grain luminescence measurements, Radiation Measurements, 37, 161-165, https://10.1016/s1350-4487(02)00170-1, 2003.

Durcan, J. A., King, G. E., and Duller, G. A. T.: DRAC: Dose Rate and Age Calculator for trapped charge dating, Quaternary Geochronology, 28, 54-61, https://10.1016/j.quageo.2015.03.012, 2015.

Kang, S., Wang, X., and Lu, Y.: Quartz OSL chronology and dust accumulation rate changes since the Last Glacial at Weinan on the southeastern Chinese Loess Plateau, Boreas, 42, 815-829, https://10.1111/bor.12005, 2013.

Kang, S., Du, J., Wang, N., Dong, J., Wang, D., Wang, X., Qiang, X., and Song, Y.: Early Holocene weakening and mid- to late Holocene strengthening of the East Asian winter monsoon, Geology, 48, 1043-1047, https://10.1130/g47621.1, 2020.

Ma, J., Chen, J., Cui, Z., Zhou, W., Chen, R., and Wang, C.: Reconstruction of catastrophic outburst floods of the Diexi ancient landslide-dammed lake in the Upper Minjiang River, Eastern Tibetan Plateau, Natural Hazards, 112, 1191-1221, https://10.1007/s11069-022-05223-z, 2022.

Murray, A. S. and Wintle, A. G.: Luminescence dating of quartz using an improved single-aliquot

regenerative-dose protocol, Radiation Measurements, 32, 57-73, https://doi.org/10.1016/S1350-4487(99)00253-X, 2000.

Prescott, J. R. and Hutton, J. T.: Cosmic ray contributions to dose rates for luminescence and ESR dating: large depths and long-term time variations, Radiation Measurements, 23, 497-500, https://10.1016/1350-4487(94)90086-8, 1994.

Rees-Jones, J.: Optical dating of young sediments using fine-grain quartz, Ancient TL, 13, 9-14, 1995.

Wang, L., Yang, L., Wang, X., and Duan, L.: Discovery of huge ancient dammed lake on upstream of Minjiang River in Sichuan , China, Journal of Chengdu University of Technology (Science & Technology Edition), 32, 1-11, https://doi.org/CNKI:SUN:CDLG.0.2005-01-001, 2005.

Wang, X.: The Environment Geological Information in the Sediments of Diexi Ancient Dammed Lake on the upstream of Mingjiang River in Sichuan Province, China, Chengdu University of Technology, Chengdu, 116 pp., 2009.

Wintle, A. G. and Murray, A. S.: A review of quartz optically stimulated luminescence characteristics and their relevance in single-aliquot regeneration dating protocols, Radiation Measurements, 41, 369-391, https://10.1016/j.radmeas.2005.11.001, 2006.

Xu, H., Chen, J., Cui, Z., and Chen, R.: Sedimentary facies and depositional processes of the Diexi Ancient Dammed Lake, Upper Minjiang River, China, Sedimentary Geology, 398, https://10.1016/j.sedgeo.2019.105583, 2020.

Yang, W.: Research of Sedimentary Record in Terraces and Climate Vary in the Upper Reaches of Minjiang River, China, Chengdu University of Technology, Chengdu, 2005.

Yang, W., Zhu, L., Zheng, H., Xiang, F., Kan, A., and Luo, L.: Evoluton of a dammed palaeolake in the Quaternary Diexi basin on the upper Minjiang River, Sichuan, China, Geological Bulletin of China, 27, 605-610, https://doi.org/10.3969/j.issn.1671-2552.2008.05.003, 2008.

Yuan, D., Cheng, H., Edwards, R. L., Dykoski, C. A., Kelly, M. J., Zhang, M., Qing, J., Lin, Y., Wang, Y., Wu, J., Dorale, J. A., An, Z., and Cai, Y.: Timing, duration, and transitions of the last interglacial Asian monsoon, Science, 304, 575-578, https://10.1126/science.1091220, 2004.

Zhu, J.: A preliminary study on the upper reaches of Minjiang River Terrace, Chengdu University of Technology, Chengdu, 73 pp., 2014.

---

## Referee Report (RR1)

Comment:

The author have given a good answer to the reviewer's comments, so I suggest directly publishing it in Earth Surface Dynamics.

---

## Referee Report (RR2)

[referee-annotated manuscript omitted]

---

## Author Response (AR2)

**Earth Surface Dynamics**

**Manuscript ID: egusphere-2023-929**

**Title: Palaeo-landslide dams controlled the formation of Late Quaternary terraces in Diexi, the upper Minjiang River, eastern Tibetan Plateau**

Dear Editor and Reviewers,

We would like to express our heartfelt appreciation for your valuable and insightful comments on our manuscript. Your expertise and guidance have significantly contributed to the improvement of our work. We are grateful for the time and effort you dedicated to reviewing our research thoroughly. Your suggestions and constructive criticism have played a crucial role in enhancing our study's clarity and overall quality. We sincerely appreciate your valuable input, which has undoubtedly strengthened our manuscript. Thank you once again for your invaluable contribution.

Please find the detailed response in the attached document.

Best regards,

Xuanmei Fan on behalf of all co-authors

**The Response to Comments from Anonymous Reviewer**

| **Specific Comments** |
|---|
| **Comment 1**
Some words and phrases are mislead readers. |
| **Response 1**

Thank you for your help in modifying the words and sentences in the manuscript. I have modified them according to your suggestions, as follows:

(1) L13-15: In this paper, we investigated the geomorphology, sedimentology, *and chronology of the terraces at the Tuanjie village (seven staircases) and at the Taiping village (three staircases) in the Diex area*.

(2) L20, L310, L315: *Late Pleistocene*.

(3) L23-25: Our analysis, combined with the tectonic uplift rate, river incision rate, and high-resolution climate data, indicates that the blockage and collapse of the palaeo-dam have been a significant factor in the formation of *the river terraces in the* tectonically active mountainous *region*.

(4) L63-64: This indicates that the Diexi palaeo-dammed lake has experienced *more than* one outburst flood event.

(5) L67-69: Due to the lack of sedimentary sequence and chronological data, *further study on the evolution of palaeo-dam and the causes of terrace formation is needed*.

(6) L112-115: *The seven terrace staircases are located in Tuanjie village* (32º2′ N, 103º40′ E) are located in Tuanjie village, on the right bank of the Minjiang River, at the mouth of the Songpinggou tributary (Fig. 1c). *The three terrace staircases are in Taiping village* (32º12′13″ N, 103º45′53″ E), at the mouth of Luobogou Gully, which is 12 km upstream of the Tuanjie (Fig. 1d) (Fan et al., 2021; Wang et al., 2005).

(7) L225-226: Tunajie *Terraces have* seven staircases, Taiping *Terraces have* three staircases, all of which are based on lacustrine deposits (Fig. 4).

(8) L277-278: The sedimentary sequences of Terraces T1 and T2 are comparable to T5 and T6 of the Tuanjie *site* (Fig. 5).

(9) L263: Angular *phyllite fragments* occur in T3.

(10) L512-513: The repetitive long-term wave erosion, fluctuating along the palaeo-lake *beach*, |

resulted in the beveling and backwearing of T2 (Malatesta et al., 2021).

(11) L567: Before 32 ka, a palaeo-dam blocked the river, with its *crest* reaching 2500 m.

**Comment 2**

L82: Width of what? stream itself, or valley bottom, or valley at some level above the stream. It should be specified.

**Response 2**

Thanks for your comments. We checked it, and rewrote it on *L82*, as follows:

The width *of the valley bottom* varies from 60 to 300 m.

**Comment 3**

L102: Sediments at the river bed are not Triassic, regardless of what was the source rock.

**Response 3**

Thanks for your comment. We modified it on *L102*, as follows:

Large amounts of *Quaternary* sediments are deposited along the Songpinggou river bed.

**Comment 4**

L106-107: Unclear. If cumulative evaporation exceeds cumulative precipitation, where the water in the river comes from?

**Response 4**

Thanks for your constructive comments.

The values of the annual cumulative evaporation, and the average temperature and precipitation are for the Diexi area. Diexi area has dry climate and little rainfall, and the precipitation is the lowest in the upper reaches of Minjiang River (Yang, 2005). So, the water of the river comes from the upstream.

We modified it on *L105*, as follows:

*In Diexi area, with* the strong effect of the prevailing winds, the annual cumulative evaporation can reach 1000-1800 mm (Yang, 2005), and the average temperature and precipitation are 13.4°C and 500-600 mm, respectively.

**Comment 5**

L196&198: which one? There are 3 terraces.

**Response 5**

Thanks for your comment. We modified it on *L196-199*, as follows:

The AMS [14]C sample collected from the *highest lacustrine deposits* of the Taiping *village* was used for comparison with the OSL sample (TP19-1), which was taken from the same position. The AMS [14]C sample collected from the *highest lacustrine deposits* of the Tuanjie *village* was compared with the AMS [14]C dating of the *highest lacustrine deposits* of the Taiping *village*.

**Comment 6**

L260-261: Not very clear what the term "strength" means here.

**Response 6**

Thanks for your comment. We modified it on *L260-261,* as follows:

These features suggest that the gravel units of T2 and T3 are clast-rich debris flows with high strength *energy* or pseudoplastic debris flows with low strength *energy*.

**Comment 7**

Line 290: blocking events that occurred upstream or downstream?

**Response 7**

Thanks for your comments. We modified it on *L290-291,* as follows:

Furthermore, the two mud-phyllite clasts layers in Taiping T3, indicate that *two blocking events occurred downstream*.

**Comment 8**

L318: Generally, paleosol, as well as modern soil is not deposited - it evolves on the already existing substrate (see section 22A.3.2.8 of McCalpin's Paleoseismology.)

**Response 8**

Thanks for your comments. We modified it on *L317,* as follows:

The ages of the paleosol of each terrace differ, but most paleosol units were *developed* during the Holocene.

**Comment 9**

L323-324: To my knowledge there landforms were mainly not deposited but eroded in lake sediments. Only thin gravel layers aere deposited over them.

**Response 9**

Thanks for your comments. We modified it on *L323,* as follows:

All the terraces were *formed* during the Holocene.

**Comment 10**

L342: Unclear how gravel units overlying lake sediments in which terraces had been eroded could be older than lacustrine silt.

**Response 10**

Thanks for your valuable comments. We rewrote it on *L341-342,* as follows:

Comparing all the ages within the Tuanjie Terraces, the gravel units *of T2 and T5* have older ages *than the lacustrine deposits of T2 and T5, respectively* (Fig. 5).

**Comment 11**

L360: Add the level (altitude) range of these terraces above the riverbed.

**Response 11**

Thanks for your valuable comments. We added the height above the river for each terrace age sample in the Supplementary. But unfortunately, not all samples have this data, we use '-' to represent no data.

**Comment 12**

L361-362: Unclear. It seems that you mix number of terraces and number of sites.

**Response 12**

Thanks for your comments. Sorry, I mixed the concepts of terrace and site. We rewrote it on *L360-363,* as follows:

Along the upper Minjiang River, there are a minimum of *fifteen terraces*, with *nine* terraces located upstream of the Diexi area (from Zhangla to Gonggaling), *two terraces* near the Diexi area *(Taiping, and Tuanjie)*, and *four terraces* are developed downstream (from Maoxian-Wenchuan).

**Comment 13**

L360-384: I think some comment will be useful how terraces described here are related with terraces eroded in the sediments accumulated in the Paleo-lake.

**Response 13**

Thanks for your valuable comments.

The formation of terraces is mainly caused by tectonic activity and climate change (Pan et al., 2003; Singh et al., 2017; Do Prado et al., 2022; Avsin et al., 2019; Gao et al., 2020), both of which have a wide range of influence. Therefore, it is necessary to describe and summarise the dating results of terraces in the upper reaches of the Minjiang River. The formation mechanism of the Diexi study area

and its upper and lower terraces is distinguished to indicate the particularity of the formation mechanism of Diexi terraces.

We rewrote it on *L376-386,* as follows:

The terraces in the area stretching from the Zhangla basin to the source of the Minjiang River are attributed to tectonic uplift (Yang et al., 2003; Yang, 2005; Yang et al., 2011; Yang et al., 2008; Chen and Li, 2014; Zhu, 2014). *Although Diexi and Zhangla are located on the Minjiang fault, in* the Diexi area, the formation and evolution of the Tuanjie and Taiping Terraces *are different, they* were influenced by the evolution of a palaeo-dam (Duan et al., 2002; Wang et al., 2005; Wang, 2009; Zhu, 2014). The *downstream* terraces in the Maoxian-Wenchuan region share similar features with the terraces in Diexi, as they are also believed to have formed as a result of the outburst of a palaeo-dammed lake (Zhu, 2014). *However, the downstream terraces are located in the Maoxian-Wenchuan fault, which makes it different from the Diexi terraces in the formation process. All these indicate that the formation and evolution of Diexi terraces are independent of the upstream and downstream terraces.* In the following sections, we will present additional evidence to explore this phenomenon further.

| **Comment 14** |
| --- |
| L417-418: Lake and dam boundary cannot be deposited. Needs rewriting. |

| **Response 14** |
| --- |
| Thanks for your valuable comments. We rewrote it on *L419-420,* as follows:

 The *dating results of the boundary of palaeo-lake* and palaeo-dam in Xiaoguanzi *supported the palaeo-lake* was *formed in* 34.87±0.76 and 35.54±0.83 cal. ka BP. |

| **Comment 15** |
| --- |
| L432-433: Unclear. Landslide had to cause lake formation first. Downstream of what? |

| **Response 15** |
| --- |
| Thanks for your valuable comments. We rewrote it on *L433-434,* as follows:

 Besides, the palaeo-landslide in Manaoding occurred at 16.75±0.62 cal. ka BP (Wang et al., 2012), suggesting that *the second outburst event happened around 17 ka*. |

| **Comment 16** |
| --- |
| L463-464: Rate during what period |

**Response 16**

Thanks for your comments. We mentioned in the first sentence:

During the damming period of the Diexi palaeo-dammed lake (32-10 ka), the incision rates in these three sections ranged from 8.3-85.3 mm/yr, 13.6-198 mm/yr, and 58 mm/yr, respectively, from upstream to downstream (Table. S2).

**Comment 17**

L476-477: But it overlaps with the period of 30-15ka

**Response 17**

Thanks for your valuable comments. We rewrote it on *L476-477,* as follows:

*Later on,* during the period of 30-*10* ka, Diexi experienced *more than* ten distinguishable climatic and environmental *stages, alternating between cold and warm* (Wang et al., 2014; Wang, 2009).

**Comment 18**

L480: is it correct to differentiate climate in so closely located areas?

**Response 18**

Thanks for your comments.

Firstly, the upper reaches of the Minjiang River are located in the transition zone from the Tibetan Plateau to the Sichuan Basin. The terrain is complex and the elevation between the upper and lower reaches is very large, resulting in significant climatic differences within the region. The instrument records that the precipitation in Diexi area is less than that the upstream and downstram regions, which indicates the particularity of Diexi area. Secondly, Zhangla basin also developed terraces, we therefore find it necessary to mention the palaeoclimatic information of both places.

**Comment 19**

L481: of what? unclear

**Response 19**

Thanks for your comments. We rewrote it on *L481,* as follows:

The chronological results *of the Tuanjie and Taiping terraces* range from 32.40±2.07 ka to 3.78±0.18 ka. We compared our dating ages with the variations in the climate curves (Fig. 7).

**Comment 20**

L485-486: Please clarify - are these gravels interbeded in lake sediments or were deposited on the

terraces eroded in lake silt? If the latter - how can it be?

**Response 20**

Thanks for your comments.

Yes, these gravels were deposited on the lacustrine deposits. As we saw in the field (Figs. 2, 3).

We were also shocked to have such a sedimentary sequence. After we obtained the dating results and investigated the sedimentological and geomorphological characteristics of the Diexi area, also the downstream of the palaeo-dam, we concluded that the gravels were deposited during the palaeo-lake damming period. The upstream materials input and eroded the lacustrine deposits, formed a channel and deposited gravels.

We rewrote it on *L486-488,* as follows:

The two gravel units *(Tuanjie T2 and T5)* are older than the lacustrine deposits of the terraces in which they are *covered*, indicating that the input of materials from the upper reaches of the Minjiang River did not cease during the blockage of the palaeo-dam, resulting in the accumulation of such a thick unit of gravels.

**Comment 21**

L496: or glaciers melt?

**Response 21**

Thank you for your reminding. Around 20,000 years ago, Diexi area may have been induced to experience large-scale hillside instability, due to the melting of glaciers in the Last Glacial Period (Wang et al., 2012). Therefore, the warming climate is leading to the overtopping of the dam, and these bodies of water may have come from melting glaciers upstream.

So, we rewrote this sentence on *L497-498,* as follows:

Although the dam-break events became more frequent during the early Holocene, it is challenging to confirm whether warmer periods triggered increased rainfall *or glacier melt*, leading to the overtopping of the dam, *its breach* and the formation of terraces.

**Comment 22**

For Figure. 7, it will be useful to specify what marks are from the Tuanjie and what from the Taiping sites.

**Response 22**

Thanks for your valuable comments. We redrew the Figure 7e with solid symbols to represent Tuanjie samples and hollow symbols to represent Taiping samples.

**Comment 23**

L526-527: deposition cannot form a channel/ Erosion forms a channel.

**Response 23**

Thanks for your comments. We rewrote it on *L528,* as follows:

Gravity and density cause the material to be deposited in the Diexi palaeo-dammed lake, *which erodes to form* a channel.

**Comment 24**

L529: I think it requires some comment. Why first outburst did not evolve continuously causing complete breach. And why paleo-lake could exist for so long time? The dam was inevitably overtopped and water had to frow over it for millennia.

**Response 24**

Thanks for your comments.

Firstly, the elevation, sedimentary stratigraphic characteristics and dating results have confirmed that the Tuanjie and Taiping terraces correspond to each other (Section 5.3). Then, the evolution process of the palaeo-dam needs to consider the Taiping terraces' sedimentary characteristics and age results. Therefore, we cannot ignore the age results of the highest lacustrine deposits of the Taiping village, which are 17.15 cal. ka and 10.77 ka, respectively (Fig. 5b). It has the same elevation (2390 m) as the highest lacustrine deposits of the Tuanjie village, but there is a 10,000-year difference.

During 27-17 ka, there was no event in the study area, and the palaeo-dam was relatively stable. Therefore, we believe the river was blocked again during 27-17 ka, causing the lake surface to expand to the Taiping village.

Secondly, it can be seen from the chronological test results of the terraces at all levels of Tuanjie and Taiping that do not present a sequence of 'from highest to lowest terraces, the ages changed from older to younger', which does not support the first outburst evolving continuously, causing the complete breach.

This paragraph mainly explains how the damming and outburst events impact upstream and downstream areas. The formation and outburst of the palaeo-dam are summarised in Section 5.4.

**Comment 25**

For Fig. 9, I would suggest to add one more stage - when the paleo-lake had been silted completely and fluvial deposition took place leading to gravels over lacustrine silt.

**Response 25**

Thanks for your comments. We added a new stage and a profile in Fig. 9, and rewrote the figure name, as follows:

[Figure]

Figure 9. Model of palaeo-landslide dam driven valley landscape and terrace evolution. The palaeo-landslide blocked the river and formed a palaeo-dam, the water level rose and formed a palaeo-dammed lake. *Lacustrine deposits were eroded to form channels, and fluvial sediments were deposited in the channels. The water overflowed the palaeo-dam.* During the outburst period, the palaeo-dammed lake shrank, the lacustrine and fluvial deposits were exposed, and the palaeo-dam was cut down to form a river channel. Subsequently, through repeated damming and outbursts, the palaeo-dam completely collapsed and deposited as outburst sediments, and terraces were formed along the river. The stratigraphical sequence of the terrace is lacustrine deposits, fluvial deposits, loess, and paleosol, from bottom to top.

**Comment 26**

L568: which one?

**Response 26**

Thanks for your comments. We rewrote it on *L570,* as follows:

During this period, the lake shore receded until the *highest lacustrine deposits of the* Taiping *village* (Fig. 10c).

**Comment 27**

For Fig. 10, Do I understand correctly that T2 formed at stage b was buried and then exhumed again?

| Looks very strange. |
| --- |
| **Response 27** |
| Thanks for your comments. |
| Yes, T2 formed in Phase b. Under the continuous erosion of fluvial sediments, channels were eventually formed in lacustrine deposits, and gravels were deposited. Subsequent lacustrine sediments were covered, but loose lacustrine sediments are difficult to deposit during subsequent evolution. |

**References**

[revised manuscript text omitted]

---

## Author Response (AR3)

**Earth Surface Dynamics**

**Manuscript ID: egusphere-2023-929**

**Title: Palaeo-landslide dams controlled the formation of Late Quaternary terraces in Diexi, the upper Minjiang River, eastern Tibetan Plateau**

Dear Editor,

We would like to extend our deepest gratitude for your invaluable feedback and suggestions on our manuscript. Your expertise and thoughtful insights have made a significant impact on the refinement of our work. We truly appreciate the time and effort you dedicated to thoroughly reviewing our research and providing detailed comments. Your constructive criticism has been instrumental in enhancing the clarity, coherence, and overall quality of our manuscript. Your input has undoubtedly elevated the scholarly value of our study. We are sincerely grateful for your invaluable contribution, which has greatly enriched our research. Thank you once again for your invaluable support and guidance.

Please find the detailed response in the attached document.

Best regards,

Xuanmei Fan on behalf of all co-authors

| Specific Comments |
|---|

**Comment 1**

Language and style: There are many instances of unclear language throughout the manuscript. I recommend a thorough proofreading and editing process to improve the clarity and coherence of the writing. This includes the use of terminology (for example, see my minor comment on the term disaster below) and the use of precise language and statements (for example, why is data incomplete and why does it make studies exploratory in Line 53). The meaning of sentences like "terraces in the Diexi area are typical fluvial terraces in the upper Minjian River") are difficult to grasp for readers that are not familiar with the study area. Also, it is unclear what "the systematic study ... is incomplete" (L61) actually means. Later, you state that "climate change can affect the topgraphy of terrace staircases" (L516). I don't know how that would work unless climate change modulates geomorphic processes that ultimately lead to incision or aggradation, both of which are processes that form terraces and which are controlled by upstream processes of sediment production and downstream changes of baselevel. These are just a few examples for many instances of unclear language that pervade the current version of the manuscript.

**Response 1**

Thanks for your comments. We proofread and edited the manuscript for clarity and coherence. For example:

L53: *The upper Minjiang, for instance, displays many terrace sequences with origins that remain debated (Yang, 2005). But due to the lack of detailed sedimentological, chronological and geomorphological information, the role of extreme geomorphic events, such as landslides and outburst floods, are still being explored (Yang et al., 2003; Yang, 2005; Gao and Li, 2006; Zhu, 2014; Luo et al., 2019).*

L61: *The Diexi terraces (Fig. 1) have been examined by previous workers (Wang et al., 2005; Yang et al., 2008; Fan et al., 2019), but a systematic analysis has yet to be conducted.*

L516: This section already modified as: *(1) A fluctuating climate may be seen in terrace geometry. In papers by Mao (2011), Jiang et al. (2014), and Shi (2020), it is argued that Tuanjie T2 displays an*

*irregular sequence of ages with depth that suggest repeated fluctuations in the lake level by up to 11 m between 19 and 11 ka (Table S2). Regarding Tuanjie T1, we note the extraordinary terrace width. Following the model described by Malatesta et al. (2021), we suggest that repetitive wave erosion associated with the fluctuating lake shoreline resulted in the bevelling and back-wearing at T1, creating a very wide terrace (Fig. 8). We note some additional erosion may have occurred owing to the positioning of the Tuanjie terraces on the concave margin of the valley (Fig. 1b) where lateral fluvial erosion tends to be accentuated.*

**Comment 2**

Research objectives: I like how you state the objectives of your study in the last paragraph of the introduction. Yet, these objectives should be described in a more concise and specific way. You write that the "purpose of this paper are". Rather, you should state the objectives of your study. Then, you write that the objective is "(1) to clarify the deposition ages and sedimentary characteristics". Here, you must be more concise in describing, what you actually do, which is offering/complementing new geochronological and sedimentological constraints/evidence. Objective 2, "to reveal the blockage and outburst of the palaeo-dam" is unclear as a study objective. It rather sounds like you are offering already here an explanation for the phenomen, which you are actually going to study. Objective 3, "to explore the influences of tectonics, climate, ..." is rather an overall aim of your study, but not an immediate objective. It will be good to make a clear distinction between research objectives and the overall aim of the study. This will also allow you to generalize the relevance of your findings to answer a broader research question.

**Response 2**

Thanks for your comments. We rewrote these sentences on *L63-68* as follows:

*Here, we seek to address the unresolved questions of the origins of the Diexi terraces, including the following aims: (1) to conduct a detailed analysis of terrace sedimentology; (2) to obtain absolute depositional ages of the terraces (at Tuanjie and Taiping); and (3) to understand the evolution of the Diexi palaeo-dam since its formation at more than 35 ka (Wang etal., 2020). Our broader objective is to provide a better understanding terrace formation linked to extreme geomorphic events in mountain*

**Comment 3**

Discussion and interpretation: The discussion of the dating results seems to me overtly optimistic concerning the precision and accuracy of the OSL ages. Moreover, the interpretation misses out processes that may affect the OSL results. For example, is it possible that OSL ages are overestimated due to incomplete bleaching which is a common problem in extreme sediment transport events (de Boer et al., 2024) and high-turbidity flows (Mey et al., 2023)? Could insufficient bleaching be an explanation why dated gravel deposits are embedded between sediments with younger ages (Fig. 5)? What about the factors contributing to the reservoir effect (L353)? Are they actually relevant in the study area (e.g. upstream carbonatic rocks, evidence for subaqueous landslides, etc.?).I have also concerns about the interpretation of beveling and backwearing of terraces by lake level variations. Can there be sufficient wave action in such a lake to suffiently erode the lake shores? What about fluvial phases during which lateral erosion dominates. Wouldn't that be much more effective in eroding the terraces? Also stating that tectonic activity is not a critical factor in the evolution of Tuanjie and Taiping terraces is highly misleading (L478). Isn't tectonic activity required to actually drive high denudation rates, landslides and the formation of landslide dams, and to lead to high erosion rates upstream? Is it possible that the three processes, extreme sediment transport events, climate, and tectonics are all important, but that they just act on different time scales and with varying stochasticity?

**Response 3**

Thank you for your comments, and sharing these two papers that have expanded my understanding of the effects of extreme sediment transport events and high turbulence on OSL ages.

Our OSL is taken from (1) lacustrine sediments, which were deposited in stable depositional environments. These samples have a certain distance from the overlying gravel unit and are not affected by the deposition of these gravels. (2) gravel units, two samples were taken from the gravel units of T5 and T2 respectively. The sedimentary facies analysis shows that they were deposited in low energy environments rather than extreme events and high turbulence. (3) paleosol unit, also not

associated with any catastrophic event.

Based on the above points and our test results, we believe that bleaching is sufficient.

There are three factors that contribute to the reservoir effect, we rewrote these three points as follows:

*(1) the lower $^{14}$C-activity carbon and the atmosphere-water exchange (Deevey et al., 1954; Keaveney and Reimer, 2012; Ascough et al., 2016); (2) landslides, debris flows, or other disturbances causing surface sediments to drop into the lake, mixing older sediments with new (Counts et al., 2015; Shi, 2020); and (3) the re-deposition of older organic components, such as stored charcoal (Kaplan et al., 2002; Krivonogov et al., 2016).*

As you mentioned, lateral erosion of the fluvial phase is effective in forming the wide surfaces of Tuanjie T1. Thanks for your reminding, this manuscript lacks this consideration. We rewrote on *L356-348* as follows:

*We note some additional erosion may have occurred owing to the positioning of the Tuanjie terraces on the concave margin of the valley (Fig. 1b) where lateral fluvial erosion tends to be accentuated.*

Tectonic activity, climate changes and extreme events all great influence on the formation and evolution of terraces. According to our analysis in sections 5.4.1 and 5.4.2, tectonic activity and climate changes were not the main factors that dominated the formation and evolution of the terraces. Therefore, to clarify, we rewrote these two sections 5.4.1 and 5.4.2 as follows:

*5.4.1 Effects of tectonism on the Diexi terraces*

*The Tuanjie and Taping terrace sites are sufficiently close (12 km) to be considered subject to the equivalent tectonic forcing. In Section 5.2, we divided the upper Minjiang River into three segments: Gonggaling to Zhangla (upstream of Diexi), the Diexi area, and the Maoxian-Wenchuan area (downstream of Diexi). Since the initial damming at the Diexi palaeo-landslide, the fluvial incision rates in these three segments of the upper Minjiang is measured at 8.3–85.3 mm/yr, 13.6–198 mm/yr, and 58 mm/yr, respectively (see Table S2). In comparison, the Minshan Block (which includes the reach from Gonggaling to Maoxian) is thought to have experienced an average uplift rate of 1.5 mm/yr during the Quaternary (Zhou et al., 2000). Clearly, recent incision rates in the Diexi area have been several-times faster than the average uplift rate of the Minshan Block. This highlights the unique character of Diexi and suggests that tectonic activity is not a primary factor in the formation of the terraces.*

**5.4.2 Effects of climate changes on the Diexi terraces**

*The regional climate has undergone three transitions from cold-dry to warm-humid climate between ~ 40 and 30 ka (Zhang et al., 2009) followed by more than ten alternations of cold to warm between 30 and 10 ka (Wang, 2009; Wang et al., 2014). The terraces at Tuanjie and Taiping span the past 32 ka, so to investigate the influence of climate we examine the climate variations over the same period (Fig. 7). The four climate proxies reveal significant fluctuations from the end of the Last Glacial Maximum (LGM) to the early Holocene followed by relative stability throughout the Holocene.*

*It is tempting to speculate that warmer periods triggered wetter conditions or glacier melt leading to the overtopping of the palaeo-dam and formation of terraces; however, we cannot see any clear relationship between the age of the terraces and the climatic variations over the past 35,000 yrs (Fig. 7). Nevertheless, two important points are worth making:*

*(1) A fluctuating climate may be seen in terrace geometry. In papers by Mao (2011), Jiang et al. (2014), and Shi (2020), it is argued that Tuanjie T2 displays an irregular sequence of ages with depth that suggest repeated fluctuations in the lake level by up to 11 m between 19 and 11 ka (Table S2). Regarding Tuanjie T1, we note the extraordinary terrace width. Following the model described by Malatesta et al. (2021), we suggest that repetitive wave erosion associated with the fluctuating lake shoreline resulted in the bevelling and back-wearing at T1, creating a very wide terrace (Fig. 8). We note some additional erosion may have occurred owing to the positioning of the Tuanjie terraces on the concave margin of the valley (Fig. 1b) where lateral fluvial erosion tends to be accentuated.*

*(2) Some degree of climate control can be recognised in terms of the aeolian and weathering processes. The loess unit at Tuanjie T4 (~13.4 ± 0.1 cal. ka BP) dates to just before the Younger Dryas reflecting a cool depositional environment; loess observed at Tuanjie T3 and T2, as well as Taiping T3 and T2 suggest ages slightly younger. Most of the palaeosol units relate to the warming conditions of the early Holocene.*

*(3) The three outburst floods (~ 27 ka, ~ 17 ka and ~ 12 ka, reported at Section 5.5) in Diexi area were only happened in the climate fluctuation periods. We speculated these floods may be the result of the glacial melting. As Wang et al. (2012) mentioned that during the Last Glacial Period, the melting of glaciers triggered massive hillslopes instability, and formed palaeo-dammed lakes.*

*(4) The absent of outburst flood in the Holocene may be related to the warm and stable climate.*

**Comment 4**

L42: The term disaster is usually used if a natural event causes wide-spread damages and loss of life. We don't know much about the impact of Quaternary extreme events on humans. Thus, instead of using the term disaster, please use the term extreme events as used in the context of magnitude-frequency relations).

**Response 4**

Thanks for your comments. We used the "*extreme events*" and "*extreme geomorphic events*" instead of "disaster".

**Comment 5**

Comment 3: It is unclear to what kind of deposits you are referring to here. If sediments are deposited along the (active) river bed, they are most likely very recent as the river bed is a product of erosion, transport and deposition of recent material. Terraces straddling the river can be of a specific age, e.g. Triassic or Quaternary. Also the underlying rocks can be of some specified age. As you write it, its rather unclear, what you mean.

**Response 5**

Thanks for your comments. Through our rearrangement, this description is not the area that we discussed in the manuscript. We deleted it.

**Comment 6**

Comment 4: Is the Diexi Valley (105) the same as Diexi area (106)? If yes, please stay consistent with geographical names.

**Response 6**

Yes, the Diexi Valley is the same as Diexi area. We used "*Diexi area*" in the manuscript.

**Comment 7**

Line 116: Please be consistent in the precision at which you report coordinates. Here, you report seconds, which are most likely not required whereas in line 113 you report only degree minutes.

**Response 7**

Thanks for your comments. We deleted the seconds.

**Comment 8**

In Line 117, you write "The course of the river is a deep canyon". Please rewrite, for example, that the river has carved a deep canyon. Maybe also state how deep the canyon is. Try to make quantitative statements where possible.

**Response 8**

Thanks for your comments. We deleted this sentence, as we described on *L81-83* as:

*The Minjiang valley widens downstream, overall, varying from 60 to 300 m wide at the valley floor (Yang, 2005; Jiang et al., 2016; Ma, 2017; Zhang, 2019), and up to 3000 m deep flanked by steep hillslopes that are typically 30-35° (Zhang et al., 2011; Guo, 2018).*

**Comment 9**

Comment 14: Please be more precise what you have dated here. As the reviewer said, the lake boundary (shore?) cannot be deposited, and hence it cannot be dated. What did you date here? Shoreline ridges? If not, are you actually dating sediments derived from litoral processes?

**Response 9**

Thanks for your comments. To clarify the second and third points, we modified it on *L374-377,* as follows:

[revised manuscript text omitted]

---

## Author Response (AR4)

**Earth Surface Dynamics**

**Manuscript ID: egusphere-2023-929**

**Title: Palaeo-landslide dams controlled the formation of Late Quaternary terraces in Diexi, the upper Minjiang River, eastern Tibetan Plateau**

Dear Dr Schwanghart,

We extend our deepest gratitude for your invaluable feedback and suggestions on our manuscript. We truly appreciate the time and effort you dedicated to reviewing the manuscript and providing insightful comments. Your suggestions have significantly enhanced the quality of the manuscript. We have carefully implemented your suggestions, and as a result, the manuscript is now more robust and scientifically rigorous.

We have included six new radiocarbon dates in the manuscript. These new dates not only reinforce the reliability of the original conclusions but also provide a valuable resource for future researchers. The relevant texts, figures and table have been revised based on these new radiocarbon dates.

Thank you once again for your invaluable support and guidance.

Please find the detailed response in the attached document.

Best regards,

Xuanmei Fan on behalf of all co-authors

| Specific Comments |
|---|

**Comment 1**

A minor, yet still unclear issue, is the role of wave erosion on the formation of the extraordinary wide terrace Tunajie T1. Do you have any additional evidence, possibly from other studies, that supports your interpretation? Are lakes of similar size actually able to create waves that are capable of performing as much erosion? If not, I suggest to formulate this explanation more carefully. Also, the schematic model of Malatesta et al. 2021 assumes fluctuating sea levels. In case of a landslide dam, this would mean that the lake either changes the height of the outlet (e.g. by additional landsliding at the lake sill), or that the lake switches from open to closed conditions. Is any of these processes actually possible?

**Response 1**

Thanks for your valuable comments.

As you pointed out, the schematic model assumes a fluctuating sea level. We reconsidered our position on this issue of beveling and backwearing of terrace, T1. In the absence of actual observations to draw upon, we provide three possible explanations. Consequently, we removed Fig. 9 and we reworked this section (*L382-388*), as follows:

*Regarding Tuanjie T1, we note the extraordinary terrace width. There are three possible factors that created the very wide T1 terrace: (i) During this period, strong monsoon activity resulted in high discharges and low sediment input, leading to river incision (Malatesta and Avouac, 2018; Tian et al., 2021; Yu et al., 2021). (ii) We note some additional erosion may have occurred owing to the positioning of the Tuanjie terraces on the concave margin of the valley (Fig. 1b) where lateral fluvial erosion tends to be accentuated. (iii) As the lowest terrace, Tuanjie T1 was subjected to frequent erosion during the progressive outburst of the palaeo-dam (Phase IV to Phase VII, Fig. 9).*

**Response 2 – For the six new radiocarbon dates**

Recently, we received six new radiocarbon dates, including three samples taken from the bottom lacustrine deposits of Tuanjie T2, and three collected from the bottom lacustrine deposits of Taiping T1, T2, and T3, respectively. These data not only strengthen the reliability of the original conclusions, also provide valuable resources for future research. Therefore, we have added these new data to this

manuscript.

Note that since we only have one date of 39 ka, and there is no further evidence to support the Minjiang River blockage before 39 ka, the blockage time is still 'before 35 ka'.

Based on these new data, the manuscript has been revised as follows:

1. Texts:

(1) In Section 3.2.2, we added a description of the test purposes of these six samples on *L175-176*, as follows:

*Six samples taken from the bottom lacustrine deposits were used to determine the depositional ages of the terraces.*

(2) In Section 4.4, we added the results of these six samples on *L286-188*, as follows:

*The bottom lacustrine deposits of T2 yielded ages of ~ 34, ~ 39, and ~ 37 cal. ka BP. The depositional ages of all three bottom lacustrine samples of Taiping T1, T2, T3 are ~ 30, ~34, and ~ 30 cal. ka BP, respectively.*

(3) In Section 5.1, we rewrote the sentence of 'older age' on *L297-298*, as follows:

[revised manuscript text omitted]